# H-NS is a bacterial transposon capture protein

Charles Cooper[1], Simon Legood[1], Rachel L. Wheat[1], David Forrest [1],
Prateek Sharma[1], James R. J. Haycocks[1] & David C. Grainger [1] ✉

The histone-like nucleoid structuring (H-NS) protein is a DNA binding factor, found in gammaproteobacteria, with functional equivalents in diverse microbes. Universally, such proteins are understood to silence transcription of horizontally acquired genes. Here, we identify transposon capture as a major overlooked function of H-NS. Using genome-scale approaches, we show that H-NS bound regions are transposition "hotspots". Since H-NS often interacts with pathogenicity islands, such targeting creates clinically relevant phenotypic diversity. For example, in *Acinetobacter baumannii*, we identify altered motility, biofilm formation, and interactions with the human immune system. Transposon capture is mediated by the DNA bridging activity of H-NS and, if absent, more ubiquitous transposition results. Consequently, transcribed and essential genes are disrupted. Hence, H-NS directs transposition to favour evolutionary outcomes useful for the host cell.

In bacteria, past horizontal gene transfer events reveal themselves as atypical sections of chromosomal DNA sequence[1]. These regions may deviate from host GC-content, use sub-optimal codons, or be conspicuously absent from close relatives[1]. Acquired by direct uptake (transformation), the action of viruses (transduction), or as transmissible plasmids (conjugation), traits encoded by horizontally acquired DNA can be beneficial[2–5]. For example, sequences encoding virulence factors, cell surface modifications and antibiotic resistance, can be obtained in this way[2,6,7]. Despite their encoded utility, the base composition of such genes can be atypical and intrinsically toxic[8]. For instance, sequences with a high AT-content are enriched for happenstance promoters of transcription[9–11]. These elements misdirect RNA polymerase and cause a global downshift in housekeeping gene expression[12]. The histone-like nucleoid structuring (H-NS) protein plays an important role by negating these effects[12,13]. Described as the genome 'sentinel', H-NS binds such loci and prevents transcription in a process known as xenogeneic silencing[13–17]. Best characterised in *Escherichia coli*, H-NS is conserved throughout the γ-proteobacteria[18,19]. In more distantly related bacteria, a variety of functionally similar proteins have been discovered[20–22]. Highlighting their intimate link with horizontal gene transfer, H-NS-like factors regulate expression of machinery for extracellular DNA uptake in diverse species[23,24].

Transposons are mobile DNA elements capable of repositioning themselves within genomes[25]. The simplest such entities are insertion sequences[26]. These encode only the machinery needed for the transposition process, often a single transposase enzyme[27]. For instance, most members of the widely studied IS5 family, found in both bacteria and archaea, possess just one open reading frame[27]. Despite their simplicity, insertion sequences can impact host cell phenotype by disrupting genes or altering their expression. For example, the horizontally acquired *bgl* operon of *E. coli*, transcriptionally silent due to binding of H-NS, can be activated by a nearby IS5 insertion[28,29]. Whilst most transposons have no intrinsic ability to move between cells, horizontal gene transfer can fortuitously deliver transposons to new hosts[30]. Indeed, IS5 was originally identified within a high AT-content section of the bacteriophage λ genome[31].

The bacterium *Acinetobacter baumannii* leads the world health organisation's list of priority pathogens that threaten human health[32,33]. Known for its ability to resist antibiotics, and opportunistically cause disease in hospital patients, *A. baumannii* has a dynamic genome subject to frequent rearrangements[34–36]. These, in part, are due to an abundance of mobile genetic elements that include insertion sequences[36]. In the present work, we identified an atypical colony variant of *A. baumannii*. Whole genome sequencing revealed insertion sequence IS*Aba*13, an IS5 family member, within the horizontally

[1]School of Biosciences, University of Birmingham, Birmingham, UK. ✉e-mail: d.grainger@bham.ac.uk

acquired K-locus bound by H-NS. Population and genome-wide mapping of ISAba13 insertion revealed transposition 'hotspots', predominantly within H-NS bound DNA. Using genomic, genetic and biochemical tools, we show that ISAba13 capture is driven by the DNA bridging activity of H-NS, not underlying DNA sequence. Consistent with this, when H-NS is absent, patterns of transposition are uniform and spatial proximity effects are revealed. We propose that, by driving transposition towards specific chromosomal regions, H-NS maximises favourable evolutionary outcomes for the host cell.

## Results

### Phenotypic diversity arising from a single transposition event in *Acinetobacter baumannii*

Hypervirulent *A. baumannii* AB5075 is a clinical strain identified in 2008[37]. We discovered atypical AB5075 colonies, grey in appearance, distinguished easily from the known translucent colony variant[38] (Fig. 1a, Fig. S1a). Strain AB5075 is known to encode two copies of ISAba13, an IS5 type transposable element, at positions 1.764 Mb and 3.863 Mb of the chromosome[39]. Using long read whole genome sequencing we found a third copy of ISAba13, specific to grey variants, located within a horizontally acquired section of the K-locus[40,41] (Fig. 1b). This AT-rich DNA region encodes enzymes needed to generate the *A. baumannii* capsule; a polysaccharide layer coating the cell. We used RNA-seq to compare wild type and grey colony transcriptomes. The presence of ISAba13 results in termination of K-locus transcription at the site of insertion (Fig. 1b). As expected, on a global scale, down regulated genes mostly correspond to the K-locus and, likely due to its increased copy number, ISAba13 appears upregulated (Fig. 1c, Table S1). Surprisingly, the biggest transcriptional effect is 45-fold down regulation of *pilA*, encoding the main structural subunit of a type IV pilus. Positioned 0.4 Mb away from the K-locus, *pilA* is genetically identical in the different cell types. Hence, altered expression is likely an indirect regulatory consequence of changes to the cell surface. Acquisition of exogenous DNA is mediated by a type IV pilus requiring *pilA*[42,43]. Hence, grey colony variants exhibit low levels of natural transformation (Fig. 1d). Phenotypically, wild type and grey colony derivatives can be distinguished in many ways. Previous work showed *A. baumannii* lacking capsule are more readily targeted by the complement component of the immune system[44]. Consistent with this, grey colony variants are more sensitive to human serum (Fig. 1e). Additionally, the grey variant is less motile and adheres to plastic surfaces to form biofilms more readily (Fig. 1f, g). As expected, clear differences in capsule production are evident (Fig. 1h). Differences in motility, biofilm formation and capsule production are quantified in Figs. S1b–d. Consistent with our observations, a parallel study, in which specific K-locus genes were deleted, also reports changes in adherence and natural transformation frequency[45].

### Mapping transposition hotspots across the *A. baumannii* genome using native Tn-seq

We were struck by the phenotypic diversity resulting from a single ISAba13 insertion. Hence, we sought to understand global and population wide dynamics of ISAba13 transposition. To do this, we developed an approach we term native Tn-seq. Briefly, standard Tn-seq exposes naive genomes to transposition in vitro, using reconstituted transpososomes, or relies on ectopic transposase expression in vivo[46,47]. The transposable element is usually semi-synthetic and defective for self-propagation. Consequently, individual cells have a similar transposon copy number and, averaged across the whole population, the genome-wide distribution of insertion events is reasonably uniform. By sharp contrast, because transposition is a rare event, most genomes in *A. baumannii* AB5075 populations will encode ISAba13 at only the 1.764 Mb and 3.863 Mb locations. In native Tn-seq, DNA amplification steps, which are unlikely to copy all instances of

ISAba13 from the starting DNA sample, are minimised to avoid further bias. When combined with sufficient sequencing depth, this allows transposition to be tracked. Using our approach, we were able to detect ISAba13 at thousands of different chromosomal locations within a culture of *A. baumannii* AB5075 cells. Strikingly, the site of ISAba13 insertion identified for the grey variant (Fig. 2a, solid blue triangle) was amongst the most commonly detected by native Tn-seq. An ISAba13 insertion previously reported by Whiteway et al. was also found but at a lower frequency (Fig. 2a, open blue triangle)[48]. Chromosome-wide, transposition is biased towards non-coding sequences (Fig. 2b). These regions account for 13% of chromosomal DNA but contain 25% of ISAba13 insertions. Within coding DNA, ISAba13 orientation was unbiased relative to the direction of the disrupted gene (Fig. 2b). The chromosome-wide pattern of ISAba13 insertions is shown in Fig. 2c as a heatmap. Darker regions indicate more frequent ISAba13 occurrence. Strikingly, most such sites have an elevated AT-content (Fig. 2c, d). Hence, the K-locus is one of many horizontally acquired AT-rich regions targeted by ISAba13.

### Transposition hotspots are DNA regions bound by H-NS

Intrigued by colocation of transposition hotspots and horizontally acquired AT-rich DNA, we determined the chromosome-wide pattern of H-NS binding using ChIP-seq (Fig. 3a, top panel). Comparison with transposition patterns revealed a strong positive correlation (r = 0.72). To understand if this relationship was causative, we repeated the native Tn-seq in the absence of H-NS. We observed a drastic rearrangement of population wide transposition; hotspots were lost in favour of more even ISAba13 distribution (Fig. 3a, b, Supplementary Data 5). Accordingly, the correlation between H-NS binding and transposition was greatly reduced (r = 0.16). Importantly, loss of H-NS did not reduce the overall number of transposition events detected (Fig. 3c). Hence, H-NS must be important for transposon capture rather than for transposition per se. We also re-examined the sequence of ISAba13 insertion sites; the strong bias towards AT-rich DNA was lost (Fig. 3d). We conclude that H-NS, rather than underlying DNA sequence, directs ISAba13 to horizontally acquired genes.

### H-NS drives transposition into plasmids and prophage

Prophage and plasmids are examples of horizontally acquired elements often bound by H-NS. This is significant because these entities can transfer transposable DNA to a new host. The *A. baumannii* AB5075 strain encodes 3 plasmids, of 83.61 kb, 8.73 kb and 1.97 kb in size and 11 prophage[37,49]. To understand the effects of H-NS, we examined our ChIP-seq and native Tn-seq data. Two of the plasmids, and 7 of the prophage, contain regions of elevated transposition dependent on H-NS binding (Table S2). In the plasmids, targeted genes encode functions related to restriction modification systems and the transport of toxic molecules. Disrupted prophage genes were also implicated in restriction modification, as well as encoding penicillin resistance and NTPase activity. We suggest that, by targeting transposable elements to prophage and plasmids, H-NS must play a key role in the evolution and propagation of mobile genetic entities.

### H-NS protects essential and housekeeping genes from transposition

None of the DNA regions bound by H-NS correspond to essential *A. baumannii* genes[50]. Moreover, H-NS bound DNA is usually transcribed poorly unless specifically induced, suggesting H-NS bound genes are not used for housekeeping functions[51]. We reasoned that, by sequestering ISAba13, H-NS may protect transcribed and essential genes from deleterious transposition events. To explore this possibility, we determined the number of ISAba13 insertion events within H-NS bound, H-NS unbound, essential and transcribed genes, in the presence and absence of H-NS. The results of the analysis are shown in Fig. S2. As expected, loss of H-NS caused a substantial decrease in

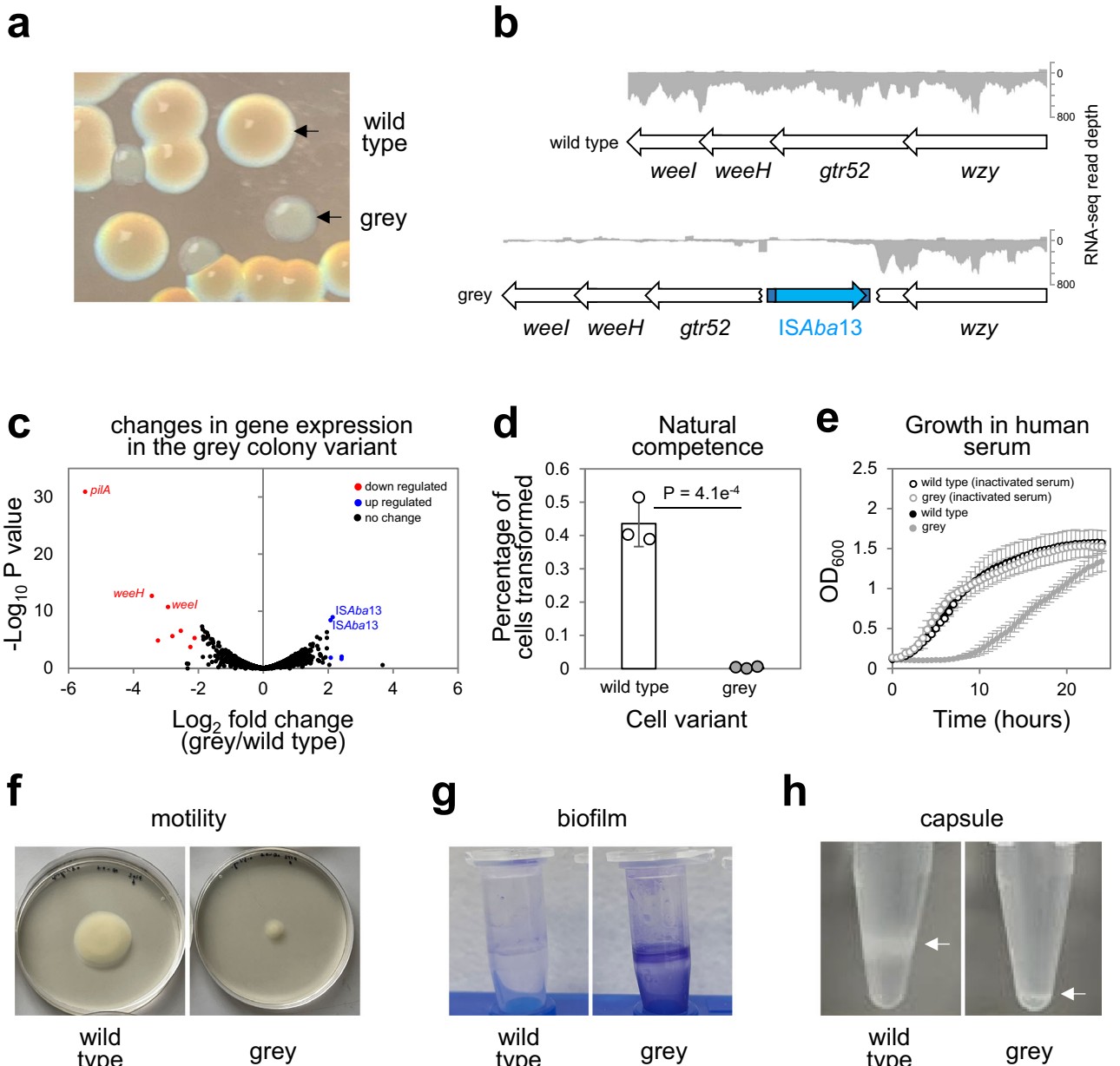

**Fig. 1 | Phenotypic diversity resulting from IS*Aba*13 transposition into the *Acinetobacter baumannii* K-locus. a** Identification of a grey colony derivative. The image shows colonies of *A. baumannii* AB5075 grown on LB agar plates. The grey colony derivative arises spontaneously. **b** Genetic basis for the grey colony phenotype. Grey colony derivatives have a copy of the insertion sequence IS*Aba*13 within the *gtr52* gene of the capsule locus. Genes are shown as arrows and labelled. IS*Aba*13 is in blue. The darker blue sections of IS*Aba*13 represent the terminal inverted repeat sequences. The traces show Illumina sequencing read depths from RNA-seq experiments. Reads mapping to the top and bottom strands of the DNA are shown as positive and negative values respectively. The *weeH* and *weeI* genes are also referred to by the pseudonyms *itrA1* and *qhbA* respectively[40,94,95]. Note that only a section of the K-locus is illustrated, the full region occupies genomic positions 3,907,256 to 3,931,673. **c** Disruption of the K-locus also causes down regulation of the *pilA* gene. The volcano plot illustrates the results of RNA-seq experiments comparing the transcriptomes of wild type and grey colony types. Each data point represents a gene and significantly up- and down-regulated genes ($P < 0.05$ and a Log2 fold change of 2 or more) are highlighted blue and red respectively. P was calculated using an exact test. Notable genes are labelled. Note that the apparent up regulation of IS*Aba*13 is likely to be an artefact due to the increased copy number of the element in grey cell derivatives. Two biological replicates were done. Source data are provided as a Source Data file. **d** Grey variants are defective for natural transformation. The bar chart shows the percentage of wild type and grey variant cells transformed by exogenous DNA. Results are the average of three biological replicates and error bars indicate standard deviation from the mean. Individual measurements are shown as discrete datapoints. A two-tailed student's *t* test was used to determine P. Source data are provided as a Source Data file. **e** Grey variants cannot grow in human serum. The graph shows growth of wild type *A. baumannii* and the grey variant in the presence of 50% (*v/v*) human serum or heat inactivated human serum. Results are the average of three biological replicates and error bars indicate standard deviation from the mean. There is a significant difference in growth for wild type and grey variants in human serum ($P = 9.3e^{-8}$) but not heat inactivated serum ($P = 0.97$) as determined using a one-way ANOVA test. Source data are provided as a Source Data file. **f** Grey colony variants are less motile. The image shows spread of wild type and grey colony variants on soft agar plates. **g** Grey colony variants better adhere to surfaces and form biofilms. The image shows crystal violet stained *A. baumannii* biofilms formed on the surface of microfuge tubes. **h** Grey colony derivatives are defective for capsule production. The images show cells separated by centrifugation in 125 µl of 30% (*w/v*) colloidal silica. Cells are visible as white bands and those with less capsule are more mobile during centrifugation.

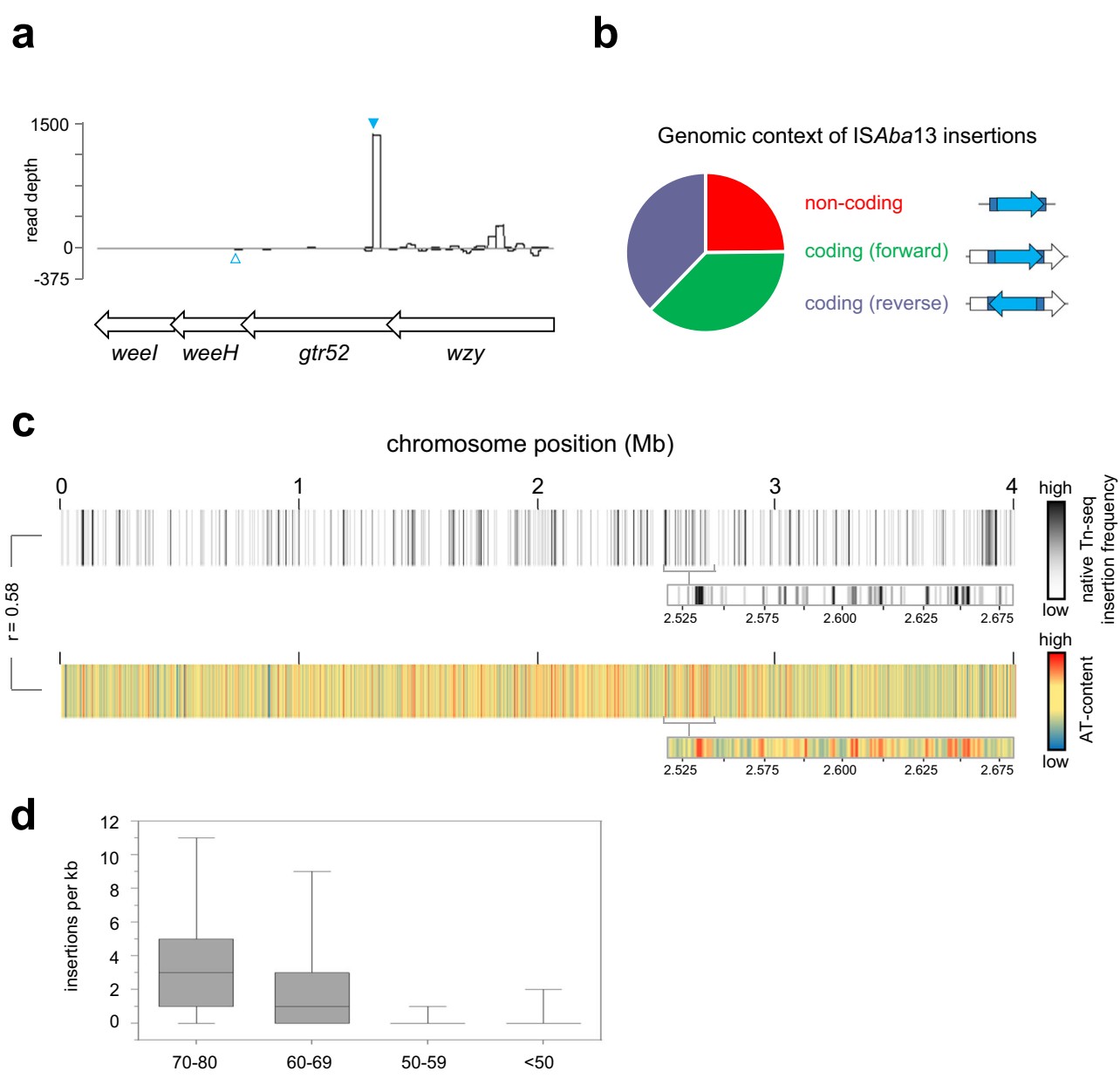

**Fig. 2 | Mapping genome and population wide transposition of IS*Aba*13 reveals a strong preference for the K-locus and other AT-rich DNA regions. a** IS*Aba*13 insertions at the K-locus. The schematic shows a section of the K-locus and corresponding read depths from native Tn-seq experiments. Genes are shown as arrows and sequence reads corresponding to IS*Aba*13 insertion in the forward or reverse orientation are given as positive or negative read depths respectively. The filled cyan triangle indicates the site of IS*Aba1*3 insertion in the grey variant described in this work. The open cyan triangle indicates the site of IS*Aba*13 insertion previously described by Whiteway et al. [48]. **b** Genomic context of IS*Aba*13 insertions. The pie chart illustrates the relative number of IS*Aba*13 insertions detected in non-coding DNA or inside genes in each possible orientation. Note that non-coding DNA accounts for 13% of the *A. baumannii* chromosome but contains 25% of the IS*Aba*13 insertions. **c** Chromosome-wide patterns of IS*Aba*13 transposition. The panel shows two heatmaps, each representing the *A. baumannii* chromosome divided into 1 kb sections. Each section is coloured according to the number of IS*Aba*13 insertions detected using native Tn-seq (top) or average DNA AT-content (bottom). The heatmap expansions are provided to aid comparison of the insertion frequency and AT-content. The Pearson correlation coefficient (*r*) of the two datasets is shown. **d** IS*Aba*13 targets AT-rich sections of the *A. baumannii* chromosome. Box plot showing the distribution of IS*Aba*13 insertion frequencies for 1 kb regions, with different AT-content, across two biological replicates. Boxes indicate the 25th–75th percentile. Horizontal lines indicate the median value. The whiskers indicate minimum and maximum values. Source data are provided as a Source Data file.

IS*Aba*13 transposition within genes usually bound by H-NS. Concomitantly, IS*Aba*13 insertions within H-NS unbound genes substantially increased, including those that are actively transcribed in the conditions of the experiment. As noted above, at a population level, IS*Aba*13 is difficult to track because transposition is a rare event. Such difficulties should be exaggerated for essential genes. Briefly, lethal IS*Aba*13 insertions will not be propagated by cell division. Hence, they

will be poorly represented in genomic DNA extracts used to make libraries for sequencing. Consistent with our native Tn-seq method being sufficiently sensitive, and able to detect signals from dead and non-dividing cells, we were able to identify IS*Aba*13 within essential genes, albeit at comparatively low levels. As predicted, in the absence of H-NS, we identified significantly more instances of IS*Aba*13 within essential DNA.

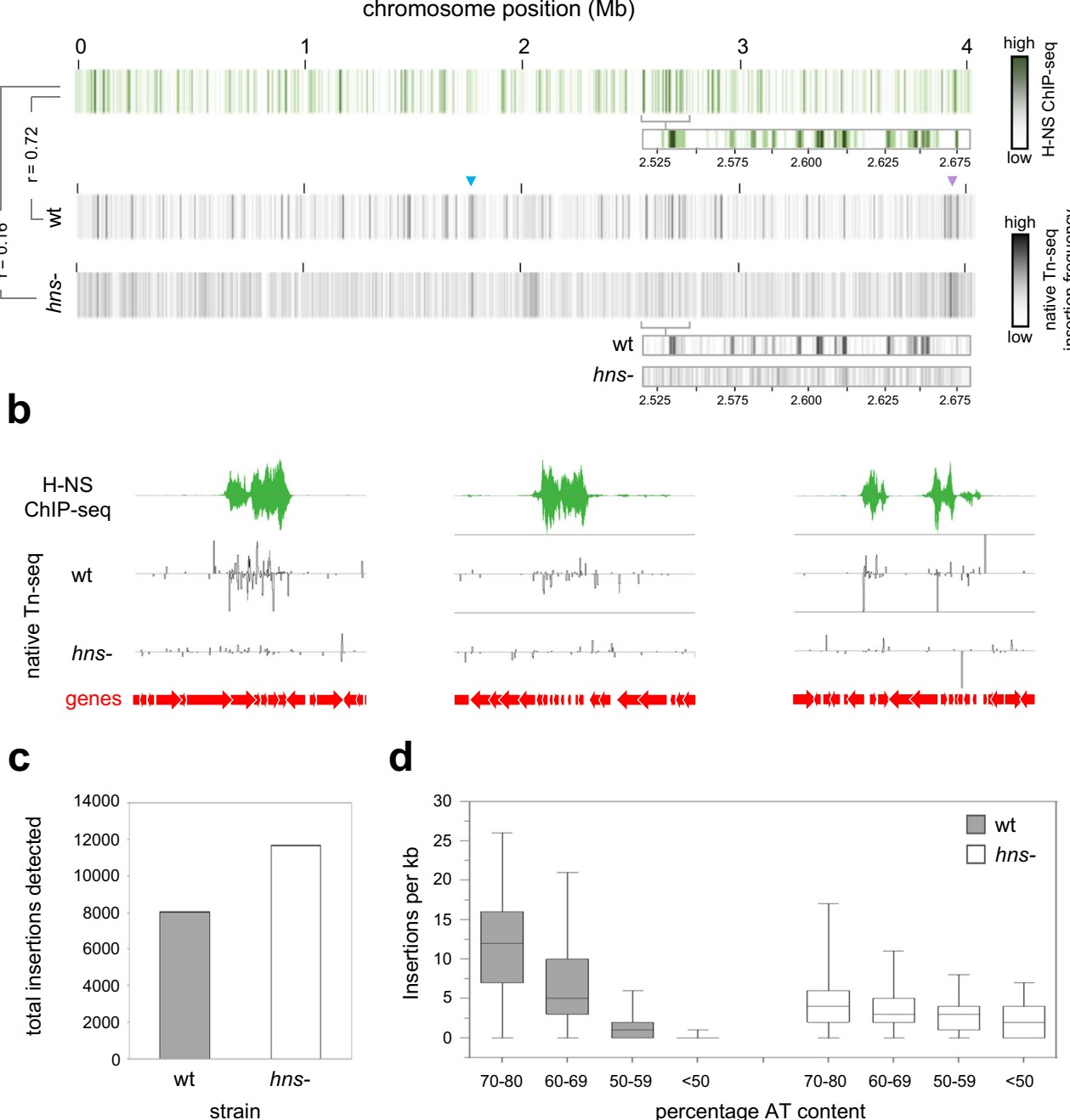

**Fig. 3 | Targeting of ISAba13 to AT-rich DNA is driven by H-NS. a** Global patterns of H-NS binding and ISAba13 transposition are correlated and H-NS dependent. The panel shows three heatmaps, each representing the *A. baumannii* chromosome divided into 1 kb sections. Sections are coloured according to the H-NS ChIP-seq binding signal (top) or the number of ISAba13 insertions detected using native Tn-seq for wild type (middle) or *hns-* cells (bottom). The heatmap expansions are provided to aid comparison of H-NS binding and insertion frequency. Pearson correlation coefficients (*r*) between datasets are shown. The raw H-NS ChIP-seq read depths are provided in Supplementary Data 1. **b** Examples of H-NS mediated ISAba13 capture. Selected chromosomal regions with H-NS ChIP-seq and native Tn-seq data shown. In both cases, traces indicate read depths with positive and negative values corresponding to the top and bottom DNA strand respectively.

Genes are shown as arrows. From left to right, the genomic loci are centred approximately around positions 134,000, 2,536,000 and 2,455,000 of the genome. **c** The number of detected transposition events is similar in the presence and absence of H-NS. The bar chart shows the number of ISAba13 insertions detected by native Tn-seq in wild type and *hns-* cells. For each strain, the number of insertions indicated is the combined total from two biological replicates. Source data are provided as a Source Data file. **d** Targeting of ISAba13 to AT-rich DNA is H-NS dependent. Box plot showing the distribution of ISAba13 insertion frequencies for 1 kb regions, with different AT-contents, in two biological replicates of wild type and *hns-* cells. Boxes indicate the 25th–75th percentile. Horizontal lines indicate the median value. The whiskers indicate minimum and maximum values. Source data are provided as a Source Data file.

## H-NS-39 inhibits H-NS mediated DNA bridging but not DNA binding

We next turned our attention to better understanding how H-NS captures transposable DNA. In this regard, the structures that H-NS forms with nucleic acids are likely to be important. It is established that H-NS can form either extended nucleoprotein filaments, with a continuous DNA tract, or bridges between separate DNA regions[52–55]. In recent work, with *E. coli* H-NS, van der Valk and co-workers showed that a section of the H-NS protein, corresponding to the dimer-dimer interaction surface, was a potent inhibitor of DNA bridging[56]. Conversely, there was little impact on DNA binding in the form of extended nucleoprotein filaments[56]. We reasoned we may be able to use a similar 39 amino acid dimer-dimer interaction region of *A. baumannii* H-NS (here on referred to as H-NS-39, Fig. S4) to specifically hinder DNA bridging. To measure bridging in vitro we used a modified version of the assay described by van der Valk et al.[56]. Briefly, radiolabelled nucleic acid is recovered with a biotin-tagged DNA sequence if H-NS forms bridges between the molecules. Hence, we used a radiolabelled section of chromosomal DNA that binds H-NS and biotinylated IS*Aba*13. Figure S3 shows the in vivo H-NS binding and transposition patterns at the respective loci. The in vitro DNA bridging result is shown in Fig. 4a. As expected, radiolabelled chromosomal DNA is recovered only in the presence of biotinylated IS*Aba*13 and H-NS. This recovery is abolished by H-NS-39. Figure 4b shows an electrophoretic mobility shift assay confirming no impact of H-NS-39 on DNA binding by H-NS.

To understand the effects of H-NS-39 in vivo we used ChIP-seq (to measure DNA binding) and 3C-seq (to monitor bridging). Consistent with our observations in vitro, ectopic expression of H-NS-39 had no impact on DNA binding by H-NS in vivo; the ChIP-seq profiles with and without H-NS-39 were almost identical (r = 0.99, Fig. 4c, Supplementary Data 1). Whole chromosome 3C-seq contact maps are shown in Fig. 4d, with interactions grouped in 10 kb bins. Overall, and similar to other bacteria, the contact map is dominated by a strong diagonal line. This results from frequent interactions between regions close in the primary DNA sequence[57,58]. Off the main diagonal, patterns of more frequent interaction manifest as squares, indicative of chromosomal interaction domains (CIDs). Hence, regions of DNA within CIDs are more likely to interact with each other than DNA loci outside of the CID. Lastly, a weaker set of interactions form a secondary diagonal line, perpendicular to the main diagonal, indicating interactions between opposing chromosomal arms (i.e. contacts between sites equidistant from the DNA replication origin in each replichore). No obvious differences are induced by H-NS-39. To confirm this, we determined fold changes in normalised contact frequencies, induced by H-NS-39, for each 10 kb bin. As expected, the resulting map is almost featureless (Fig. S5a). Previous chromosome conformation capture studies in *E. coli* also reported minimal effects of H-NS on 10 kb resolution contact maps[57]. This is likely because H-NS mediated interactions occur between chromosomal loci separated by <10 kb. Hence, we grouped contacts in 1 kb bins and again calculated fold changes induced by H-NS-39. At this resolution, differences are clear (Fig. S5b). Most notably, short range interactions are reduced and these changes are evident chromosome wide (see red diagonal signal in Fig. S5b). Consistent with this, differences in local DNA folding patterns are evident upon visual inspection of the 1 kb 3C-seq matrices (Supplementary Data 2–4). For instance, we identified signals indicative of loops that are lost when H-NS-39 is expressed. An example is shown in Fig. 4e, alongside ChIP-seq data for H-NS binding. Further examples are shown in Fig. S5c–f. Importantly, whilst the loss of DNA loops is a prominent difference, more subtle changes to short range interactions are near ubiquitous (compare the entirety of the plots in Fig. 4e and S5c–f). Most likely, some of these changes are an indirect consequence of DNA bridging by H-NS being altered.

## H-NS mediated DNA bridging drives transposon capture

Our observations are consistent with H-NS-39 impeding the ability of H-NS to bridge between, rather than bind to, individual DNA sections in the conditions used. We next wanted to understand the impact of H-NS-39 on patterns of IS*Aba*13 transposition. The results of native Tn-seq analyses, with or without H-NS-39 expression, are shown in Fig. 5a and Supplementary Data 5. As expected, in the absence of H-NS-39, transposition hotspots closely coincide with DNA regions bound by H-NS (r = 0.68). When H-NS-39 is expressed, transposition events are more uniformly distributed, similar to transposition patterns observed in cells lacking H-NS completely (compare Figs. 3a and 5a). Hence, the degree of correlation between H-NS binding and transposition frequency was reduced (r = 0.35). Specific examples of H-NS binding, and the impact of H-NS-39 on transposition, are shown in Fig. 5b. We conclude that DNA bridging is a key aspect of IS*Aba*13 capture by H-NS bound DNA.

## Transposition events are likely captured after IS*Aba*13 excision

DNA bridging could support capture of IS*Aba*13 in one of two ways. First, interactions between IS*Aba*13, and an H-NS bound region, could form prior to excision of the transposable element. In this scenario, chromosomal sites containing IS*Aba*13 should interact more frequently with H-NS bound loci. Alternatively, capture of IS*Aba*13 could occur after excision of the element. In this scenario, the diffusing transpososome would interact with H-NS bound DNA. Hence, increased contact frequency between H-NS bound regions, and chromosomal IS*Aba*13 sites, would not be required. To differentiate between the models, we re-examined our 3C-seq data. First, we divided the chromosome into 1 kb bins and determined the number of contacts involving each bin. Next, we ranked the bins according to the number of 3C-seq contacts. The results of the analysis are shown in Fig. S6a. Strikingly, bins containing each copy of IS*Aba*13 are amongst the most poorly interactive sections of the chromosome. Furthermore, expression of H-NS-39 had little impact on the ability of these regions to interact with other parts of the DNA. Second, we selected all contacts involving IS*Aba*13 at positions 1.764 Mb or 3.863 Mb of the chromosome. The bar chart in Fig. S6b shows the number and location of contacts made by each IS*Aba*13 copy. In both cases, most interactions occur within a ~20 kb region surrounding the element; contacts with chromosomal regions elsewhere are both infrequent and uniform (Fig. S6b). This contrasts sharply with transposition frequency (Fig. S6c). Hence, DNA loci containing IS*Aba*13 do not interact with distant H-NS bound transposition hotspots. This suggests that H-NS must capture IS*Aba*13 after excision from the chromosome. However, proximity to an existing copy of IS*Aba*13 could impact transposition independently of H-NS.

## Proximity is a major driver of transposition frequency in the absence of H-NS

It is curious that sites surrounding IS*Aba*13, at chromosomal positions 1.764 Mb or 3.863 Mb, still experience high levels of transposition when H-NS is deleted (blue and purple triangles in Figs. 3a and 5a). To better understand the role of proximity, we introduced a new copy of IS*Aba*13, within *ompW* and used native Tn-seq to map transposition. The presence of IS*Aba*13 leads to a broad area of increased transposition surrounding *ompW* (Fig. 5c). This suggests immediate proximity of a chromosomal site, to a copy of IS*Aba*13, is indeed important. We reasoned that, in the absence of H-NS, it might be possible to detect subtle proximity effects normally obscured when H-NS is present. Our 3C-seq analyses show that each arm of the *A. baumannii* chromosome interacts with the other (Fig. 4d). Thus, we considered the possibility that transposition frequencies might correlate at equivalent sites on each chromosomal arm. To test this, we divided each chromosomal arm into twenty 100 kb bins (illustrated in Fig. 5d). We then

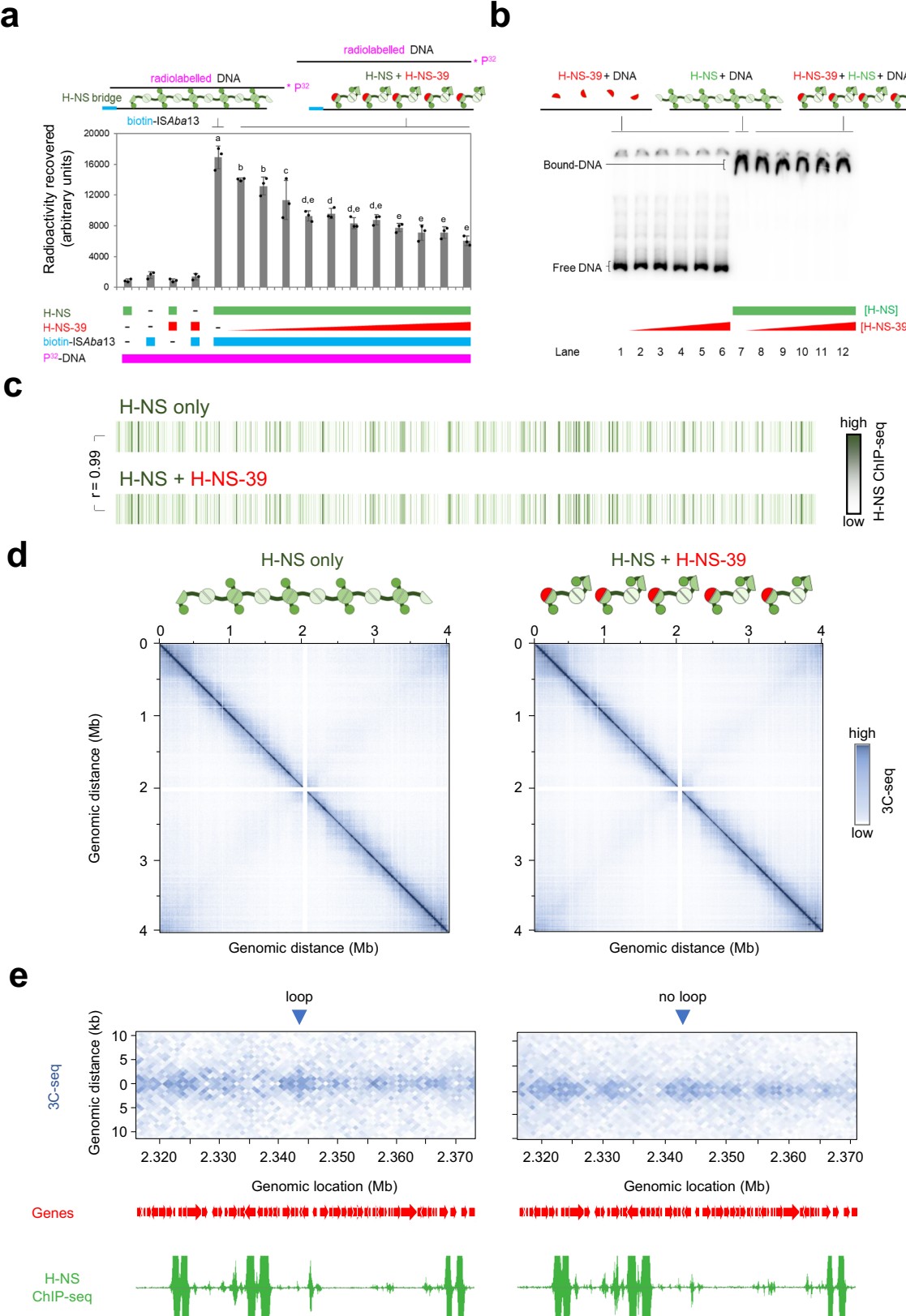

determined the number of transposition events, in the presence and absence of H-NS, for corresponding bins on each arm of the chromosome. The data for wild type cells are shown in Fig. 5e, where each datapoint corresponds to one of the bin pairs in Fig. 5d. We observed a weak positive correlation between transposition frequencies at equivalent locations in each replichore (r = 0.31).

Strikingly, in the absence of H-NS, the correlation increased (r = 0.62). We conclude that proximity to an existing ISAba13 copy can increase the likelihood of transposition to a given site. However, except when chromosomal sites are immediately adjacent to ISAba13 (Fig. 5c) these effects are largely hidden by the influence of H-NS.

**Fig. 4 | The H-NS multimerization region blocks H-NS mediated DNA bridging in vitro and in vivo. a** H-NS-39 interferes with H-NS mediated DNA bridging in vitro. The bar chart illustrates the amount of radiolabelled DNA recovered by H-NS mediated bridging interactions with biotin labelled IS*Aba*13. The radiolabelled sequence corresponds to the H-NS bound regulatory DNA upstream of genes encoding the type 6 secretion system of *A. baumannii* (Fig. S3a). Where present, H-NS was used at a final concentration of 6 µM. The bridging interaction is disrupted by H-NS-39, added at final concentrations between 0.5 and 10 µM. Results are the average of three independent experiments and error bars indicate standard deviation from the mean. The schematic above the graph illustrates the procedure. DNA fragments are shown as solid lines and H-NS is in green. Individual H-NS molecules possess surfaces for DNA binding (circle), dimerisation (dark green semi-circle) and multimerization (pale green semi-circle). H-NS-39 (red) consists of the multimerization surface only. Different letters above bars indicate significantly different groups according to an unpaired one-way ANOVA with Tukey's HSD test ($P = 6.63e^{-11}$). Source data are provided as a Source Data file. **b** H-NS-39 does not bind DNA or interfere with binding of H-NS to DNA in vitro. Results of an electrophoretic mobility shift assay to measure binding of H-NS to the regulatory region of genes encoding the type 6 secretion system of *A. baumannii*, and the impact of H-NS-39. Where present, H-NS and H-NS-39 were used at final concentrations of 6 µM and 1–10 µM respectively. The schematic is as described for (**a**). The experiment

was done twice. **c** Global DNA binding by H-NS in vivo is the same in the presence and absence of H-NS-39. The panel shows two heatmaps, each representing the *A. baumannii* chromosome divided into 1 kb sections. Sections are coloured according to the H-NS ChIP-seq binding signal in the absence (top) or presence (bottom) of ectopic H-NS-39 expression. The Pearson correlation coefficient (*r*) between datasets is shown. The raw H-NS ChIP-seq read depths are provided in Supplementary Data 1. **d** Global 10 kb resolution 3C-seq contact maps are the same in the presence and absence of H-NS-39. The heatmaps illustrate interaction frequencies between 10 kb sections of the *A. baumannii* chromosome, measured by 3C-seq, in the presence and absence of H-NS-39. Axes indicate the genomic location of each bin in the pair. Individual sections are coloured according to the number of interactions between the two corresponding chromosomal locations. The contact matrix values, and explorable versions of each matrix, are in Supplementary Data files 2–4. **e** H-NS-39 alters short range interactions in 1 kb resolution 3C-seq contact maps. The heatmaps illustrate interaction frequencies between 1 kb sections of the *A. baumannii* chromosome, measured by 3C-seq, in the presence and absence of H-NS-39. An interaction pattern indicative of a loop is marked by a blue triangle. Signal in this region is lost in the presence of H-NS-39. The locations of genes (red arrows) are also shown alongside H-NS binding patterns determined by ChIP-seq with or without expression of H-NS-39. The contact matrix values, and explorable versions of each matrix, are in Supplementary Data files 2–4.

## H-NS is a transposon capture protein in *Escherichia coli*

Prior to genome-scale approaches being commonplace, several groups investigated interplay between H-NS and transposition[59–62]. Most notably, Swingle et al. reported more uniform distribution of IS*903* in *E. coli* cells lacking H-NS[59]. In the absence of information about global H-NS binding, the authors reasoned that IS*903* might be excluded from H-NS bound regions. Their logic was that such sites would become available for transposition upon H-NS deletion, accounting for more even transposition. Given our observations for *A. baumannii*, and with the benefit of genome-wide H-NS binding data for *E. coli*[63], we reassessed the transposition patterns defined by Swingle et al. [59]. In wild type *E. coli*, 19 of 32 IS*903* insertions fell within DNA bound by H-NS, encompassing 8 chromosomal regions with multiple insertions defined as hotspots. In the absence of H-NS, none of these hotspots were detected and only 11 of 33 IS*903* insertions were within H-NS bound regions. This resembles chance coincidence with H-NS binding; 10 of 33 random locations were H-NS associated. Hence, the findings of Swingle et al., for IS*903* in *E. coli*, are consistent with our observations here, for IS*Aba*13 in *A. baumannii*. For further confirmation, we mapped the distribution of an additional *E. coli* insertion sequence, *insH3*, across a population of *E. coli* cells. As expected, *insH3* insertion hotspots located to H-NS bound DNA (Fig. S7). Note that, like IS*Aba*13, both *IS903* and *insH3* are in the IS5 family.

## Discussion

H-NS is widely understood to inhibit transcription of horizontally acquired AT-rich genes by virtue of its ability to form bridges and filaments with DNA[18,64]. Primarily, this prevents spurious intragenic transcription initiation that otherwise diverts RNA polymerase from housekeeping functions[9,11,12]. At some loci, H-NS is also integral to the regulatory strategy for mRNA production; anti-silencing mechanisms allow gene expression in response to environmental cues[51]. Hence, the consensus view, that H-NS is a 'sentinel' managing the transcription of horizontally acquired DNA, is widespread[13]. In this work, we have identified transposon capture as a major overlooked H-NS function. This finding is counterintuitive, because H-NS is generally considered a protein that excludes other factors from bound DNA[65–67]. Indeed, in their prior work, Swingle et al. assumed H-NS might block access of IS*903* to parts of the *E. coli* genome[59]. Comparison to recent H-NS binding data shows that Swingle and colleagues had in fact observed the transposon capture activity of H-NS reported here. By contrast, Kimura and colleagues showed exclusion of mariner transposon from H-NS bound regions of the *Vibrio cholerae* chromosome[68]. The mariner transposon is eukaryotic in origin, not adapted to bacterial

nucleoprotein, and absolutely requires a TA insertion site that will be hidden in H-NS bound DNA. As such, any relevance to prokaryotic transposition is unclear. Even so, we caution that H-NS may not influence all transposable elements. For instance, we demonstrate that DNA bridging by H-NS is required to capture IS*Aba*13. It is possible that some insertion sequences do not engage in such interactions.

Transposon capture by H-NS has implications for understanding the dissemination and diversification of other mobile genetic entities. As discussed above, plasmids and prophage are often AT-rich, or contain sections of high AT-content DNA. Consistent with this, we detect H-NS dependent transposition hotspots within both prophage and plasmids. Similarly, two of the hotspots identified by Swingle et al. correspond to H-NS bound prophage DNA in *E. coli*[59,63]. Thus, in addition to controlling the transcription of horizontally acquired genes, H-NS controls their acquisition of transposable DNA. This is likely to have important downstream consequences. For instance, by facilitating transposition into prophage or plasmids, H-NS provides a route via which transposable elements can spread horizontally. In other cases, prophage or plasmid transmission could be inactivated by transposition into key genes. This may explain why phage and plasmids encode H-NS modulating factors.

In summary, H-NS manipulates transposition to maximise evolutionary outcomes for the host; important genes are protected whilst variability arises in pathogenic traits. This adds complexity to the prevailing view that most genetic change occurs at random. For *A. baumannii*, the H-NS bound K-locus frequently acquires IS*Aba*13 and this generates substantial phenotypic change (Fig. 1). In clinical environments, this likely offers the bacterium an important bet hedging strategy. Whilst loss of capsule renders the cell more sensitive to components of the immune system, the microbe is better able to adhere to surfaces and form biofilms (Fig. 1). Hence, grey derivatives may better colonise abiotic environments than human hosts. Crucially, since IS*Aba*13 insertion is reversable, and excision does not leave a DNA sequence scar, this does not represent an evolutionary dead end[48]. We note that IS*Aba*13 can target different parts of the K-locus (Fig. 2a, Supplementary Data 5). Depending on the gene disrupted, different phenotypes could emerge. For example, in hypervirulent *A. baumannii* strain HUMC1, K-locus gene *gtr6* is inactivated by IS*Aba*13[69]. Most likely, insertion acquisition was mediated by H-NS and natural selection then fixed this genetic change in the population. Rather than preventing capsule synthesis, loss of *gtr6* subtly alters capsule properties. As a consequence, strain HUMC1 is not phagocytosed by macrophages and causes lethal infections in mice[69]. Intriguingly, in many clinical isolates, IS*Aba*13 has substantially proliferated. For instance,

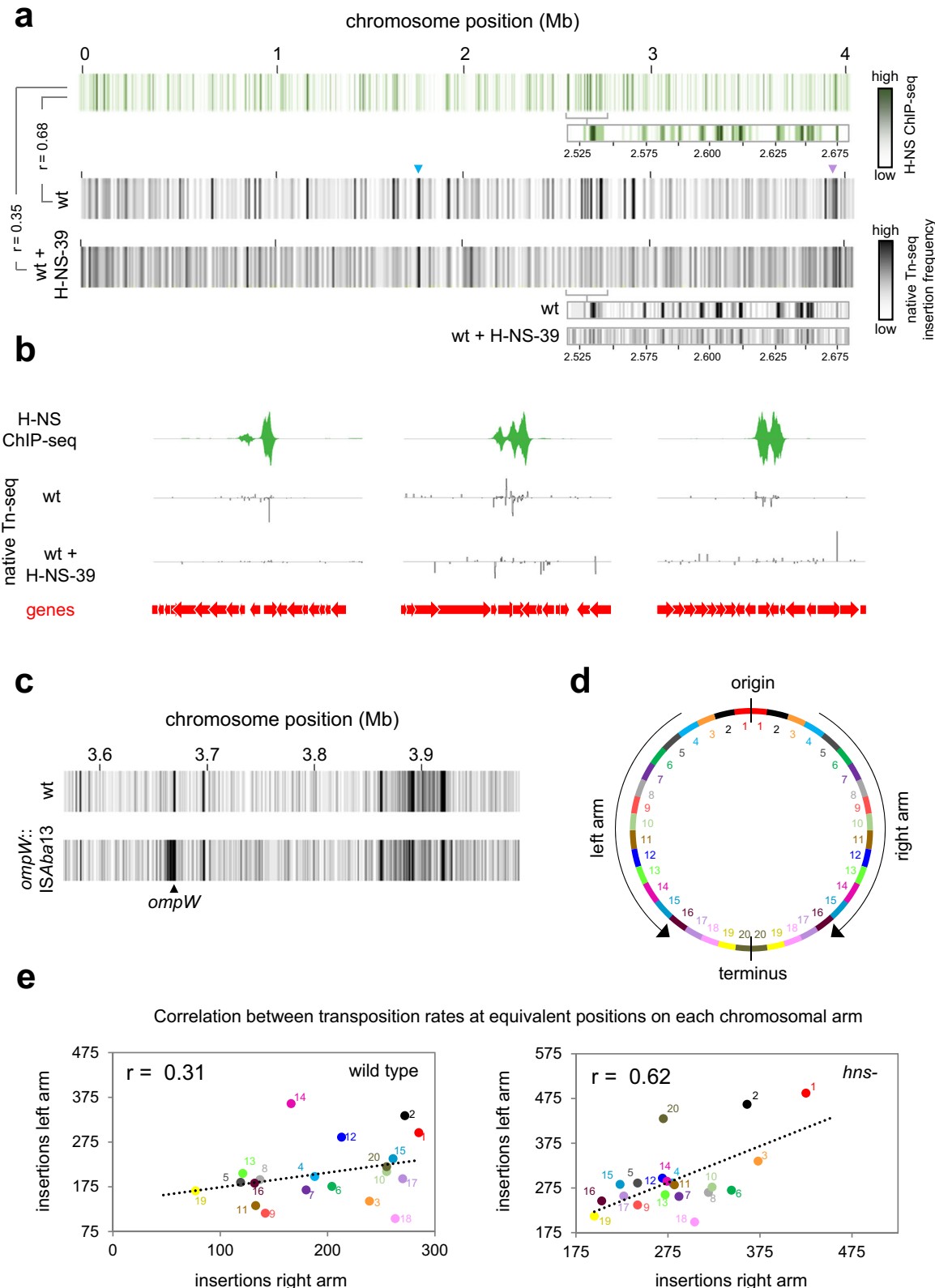

the hypervirulent LAC-4 isolate has 22 copies of the element[70]. These are predominantly found within prophage and genes encoding cell surface factors. Understanding how this modulates immunogenicity and survival in clinical settings is a key future goal. We predict that H-NS plays a major role in *A. baumannii*, by driving transposition into important pathogenicity determinants, to create useful phenotypic heterogeneity in populations.

## Methods

### Strains, plasmids and oligonucleotides

Strains, plasmids and oligonucleotides are listed in Tables S3 and S4. Unless otherwise stated, strains were grown in low-salt Lysogeny Broth (LB) medium, with shaking for liquid cultures. To avoid the use of sub-cultures, liquid media, inoculated directly from a glycerol stock, was left at room temperature overnight before being transferred to 37 °C, with

**Fig. 5 | DNA bridging is required for capture of transposition events by H-NS.**
**a** H-NS-39 causes more uniform ISAba13 transposition patterns globally. The panel shows three heatmaps, each representing the *A. baumannii* chromosome divided into 1 kb sections. Sections are coloured according to the H-NS ChIP-seq binding signal (top) or the number of ISAba13 insertions detected using native Tn-seq in the absence (middle) or presence (bottom) of H-NS-39 expression. The heatmap expansions are provided to aid comparison of H-NS binding and insertion frequency. Pearson correlation coefficients (*r*) between datasets are shown.
**b** Examples of H-NS-39 mediated changes to ISAba13 transposition patterns. Selected chromosomal regions with H-NS ChIP-seq and native Tn-seq data shown. In both cases, traces indicate read depths with positive and negative values corresponding to the top and bottom DNA strand respectively. Genes are shown as arrows. **c** Transposition hotspots arise immediately adjacent to existing copies of

ISAba13. The panel shows two heatmaps, each representing a 0.4 Mb region of the *A. baumannii* chromosome divided into 1 kb sections. Introduction of ISAba13 in *ompW* creates a transposition hotspot, around 10 kb in size, surrounding the insertion site. **d** Map of the *A. baumannii* chromosome. The schematic illustrates the *A. baumannii* chromosome divided into 40 sections, each 100 kb in length. Equivalent sections in the left and right replichores are coloured and numbered accordingly. **e** The number of detected transposition events correlate between equivalent positions on each chromosomal arm. The scatter plots show the number of ISAba13 insertions, detected by native Tn-seq, at equivalent positions on each chromosomal arm, in the presence and absence of *hns*. Data points are coloured according to the schematic in (**d**) and *r* values are correlation coefficients. Source data are provided as a Source Data file.

shaking, until cells reached the required growth phase. The exception was generation of cultures for native Tn-seq assays, where overnight subcultures, incubated at 37 °C in LB, were used. Our logic was that subculturing should help to increase the number of different transposition events that we were able to detect. To produce H-NS-39 in cells we used plasmid pVLR1Z, present at around 50 copies per cell, with expression driven by a copy of the *hns* promoter[71]. Note that the *hns*- AB5075 strain derivative is disrupted by T26 at position 211 of the gene.

### Strain construction
The *A. baumannii* chromosome was altered to encode H-NS with a C-terminal 3xFLAG fusion using the method of Godeux et al. [72]. Briefly, a DNA fragment encoding the *hns*3xFLAG fusion, the gene *sacB* and resistance to apramycin, was prepared by Gibson assembly and used to transform naturally competent *A. baumannii* AB5075. Apramycin resistant cells were transformed with a second DNA fragment, similarly prepared, encoding the *hns*3xFLAG allele but lacking *sacB* and the antibiotic resistance cassette. Transformants were isolated by counterselection with sucrose. The presence of the desired genetic change was confirmed by PCR and DNA sequencing. Finally, expression of H-NS 3xFLAG was confirmed by western blotting. To insert ISAba13 at the *ompW* locus we used a similar appraoch[72]. To begin, PCR was used to generate ~2 kb fragments flanking *ompW*. Next, two further fragments, containing ISAba13 and *sacB*-*apra*R respectively, were also generated. The different sections of DNA were then combined by Gibson assembly, purified and used to transform AB5075 cells. To remove the *sacB*-*apra*R cassette from the locus, leaving a scarless ISAba13 insertion, we first extracted genomic DNA from a successful transformant. This was used as a template to generate fragments, that could be fused by overlap extension PCR, to create a copy of the *ompW* locus lacking the *sacB*-*apra*R cassette. Following transformation, successful recombinants were plated on media containing 20% (*w/v*) sucrose for counter selection.

### Natural transformation of *A. baumannii*
To naturally transform AB5075, cells were first streaked onto LB agar and incubated overnight at 37 °C. Next, 1 ml of motility medium (2% (*w/v*) agarose and 5 g/L tryptone) was dissolved by heating and added to a sterile microfuge tube. Separately, a single AB5075 colony was resuspended in 2 ml LB and incubated at 37 °C until the $OD_{600}$ was -1. The culture was diluted 100-fold in sterile PBS and 2.5 µl mixed with an equal volume of transforming DNA. This mixture was spotted onto motility medium and incubated at 37 °C overnight. The next day, 100 µl of PBS was added to the Eppendorf tubes and cells were resuspended using a vortex mixer for 5 s. The resulting mixture was plated on appropriate media. For the assays shown in Fig. 1d, cells were transformed with 250 ng of pVLR2Z and serially diluted prior to spreading on LB agar with and without zeocin.

### Whole genome sequencing
The genomes of our *A. baumannii* AB5075 laboratory stock, and the grey colony derivative, were sequenced. Strains were grown to

mid-exponential phase and $10^9$ cells were collected by centrifugation. Cells were washed with 1 ml PBS and resuspended in 0.5 ml 1x DNA/RNA shield (Zymo). Further processing, long read genome sequencing (Oxford Nanopore) and genome assembly, was done by MicrobesNG. Briefly, Kraken was used to identify a reference genome and reads mapped using BWA mem[73,74]. A de novo assembly was performed with SPAdes and reads mapped back to contigs with BWA mem[75]. The chromosome sequences are available from GenBank using accession codes CP144559 and CP144563, for the wild type and grey colony variants respectively. The 8731 and 1967 bp plasmids have accession codes CP144560 and CP144562 respectively for wild type cells. The equivalent accession codes for the grey variant are CP144564 and CP144565. Note that, in our assemblies, only a portion of the 83.61 kb plasmid, p1AB5075, described previously (GenBank accession CP008707.1) is represented. This is likely an assembly artefact since, in Illumina sequencing experiments, we were able to map reads across the entirety of CP008707.1, albeit with a reduced depth of coverage compared to other genomic elements. This may be because the 83.61 kb plasmid is sometimes lost from cells during laboratory culture. Alternatively, the plasmid DNA could be isolated less efficiently during preparation of samples for genome sequencing.

### Capsule production assays
We followed the protocol of Kon et al. [76]. Cells were collected from an overnight culture by centrifugation, resuspended in 1 ml PBS, and 875 µl of resuspended bacteria were mixed with 125 µl of 30% (*w/v*) Ludox LS colloidal silica. Mixtures were centrifuged at 9000 × *g* for 30 min. Band positions were recorded photographically and distances measured using ImageJ.

### Bacterial motility assays
Five bacterial colonies were resuspended in 100 µl LB and 1 µl of the resulting mixture spotted onto 0.3% agarose plates. These were sealed with parafilm and incubated, agar surface up, at 37 °C for 16 h inside a sealed plastic bag. Plate images were recorded photographically, and motility was quantified by measuring the colonised area of the plate with ImageJ.

### Biofilm assays
The procedure was adapted from prior studies[77,78]. Bacteria were grown to mid-log phase and 750 µl of culture added to a 1.5 ml polypropylene microfuge tube. After incubation at room temperature for 24 h the culture was removed and tubes washed twice with 1 ml PBS. Next, 850 µl of 0.1% (*w/v*) crystal violet was added to the tube. Following incubation at room temperature for 15 min, the dye was removed and tubes were washed three times with 1 ml PBS. After drying, tubes were photographed as a qualitative measure of biofilm production. For quantification, the dye was dissolved by adding 1 ml of 100% ethanol to each tube for 10 min and followed by thorough mixing. The $A_{585}$ was recorded to measure amount of crystal violet dye.

## RNA-seq: library preparation and sequencing

Transcripts were isolated from cells using a Qiagen RNeasy kit. Library prep for RNA-seq was done by Vertis biotechnologie as described previously[79]. Samples were depleted of rRNA, fragmented ultrasonically and an adapter ligated to the 3' end. M-MLV reverse transcriptase was used to generate cDNA that was then amplified by PCR. Libraries were sequenced using an Illumina NextSeq 500 system with 75 bp read length. The raw data are available from ArrayExpress (accession code E-MTAB-13791). Experiments were done in duplicate.

## RNA-seq: bioinformatics

Sequencing reads were processed as described by Forrest et al. [79]. Briefly, reads were aligned against the *A. baumannii* AB5075 reference genome (GenBank accession CP008706.1) using Bowtie 2[80] (Galaxy Version 2.4.2) and coverage extracted using the genomcov function of BedTools[81] (Galaxy version 2.30.0). FeatureCounts (version 2.6) of the Rsubread[82] (version 2.6) package was used to determine gene read counts, which were inputted into the exact function of edgeR[83] (Galaxy version 3.34.0) to determine differential gene expression. For the analysis in Fig. S2, we then determined the average RNA-seq read depth in 50 bp windows. If the average of all windows within a given gene was >99 this was selected as transcribed. The cutoff was selected after visual comparison of transcribed and non-transcribed regions.

## Native Tn-seq: library preparation and sequencing

Our procedure was derived from the method of Long et al. [84]. Two biological replicates were done for each genetic background. Following overnight growth, 1 ml of cells were sub-cultured in 50 ml LB and grown to mid-exponential phase ($OD_{650} = 0.5$). Genomic DNA was extracted from 13 ml of the culture using a DNeasy Blood & Tissue kit (Qiagen) according to the manufacturer's instructions for Gram-negative bacteria. The optional RNase A step was included and the final DNA elution was done using 100 μl of $dH_2O$. Extracted DNA was further purified with Agencourt AMPure XP magnetic beads, using a 1.8:1 ratio of beads:DNA, according to the manufacturer's instructions. After washing DNA bound beads twice, with 200 μl of 70% (v/v) ethanol, DNA was resuspended in deionised $H_2O$. NEBNext dsDNA fragmentase was used to digest 6 μg of genomic DNA. Reactions were incubated for 10–15 min at 37 °C and stopped by addition of 5 μl 0.5 M EDTA to the 20 μl reaction mix. Fragmented DNA, between 400 and 650 bp in size, was extracted from a 1% (w/v) agarose gel following electrophoresis. Gel slices were stored at −20 °C for >30 min and DNA extracted using the Qiagen MinElute kit according to manufacturer's instructions with an extra PE buffer wash. Next, 1 μg of DNA was mixed with 7 μl NEBNext Ultra II end prep reaction buffer (NEB) and 3 μl Ultra II end prep enzyme mix (NEB), and incubated at 20 °C for 30 min, followed by 65 °C for 30 min. This creates a single adenine base overhang at the 5' end of each DNA strand.

Next, 48 μl of 100 μM stock DNA adaptor oligonucleotides 1.2 and 2.2 were mixed with 4 μl 50 mM MgCl₂, at 95 °C for 10 min and allowed to anneal by cooling to room temperature overnight. Once base paired, the longer adaptor 2.2 strand creates a 5' overhang. The recessed 3' end of adaptor 1.2 has a C3-amino modification to block strand extension during PCR. Importantly, there is also a single thymine base overhang at the 3' end of adaptor 2.2. This facilitates ligation to genomic DNA fragments. The annealed adapter was attached to DNA fragments using either a Fastlink DNA ligation kit (Cambio) or, following discontinuation of this reagent, the NEBNext ultra II ligation module (NEB). After ligation, products were purified with a 0.7:1 ratio of AMPure XP beads to DNA. Purified products were used as template for 50 μl PCR reactions containing 1x GoTaq (Promega), 0.3 μM each of oligonucleotides JelAP1 and either ISAba13_out or insH3_out and 5% (w/v) DMSO. For each experiment, 8 reactions were done using 100 ng of DNA template in each. The PCR conditions comprised an initial 95 °C denaturation step, followed by 20 cycles of

95 °C (30 s), 58 °C (30 s) and 72 °C (45 s) and a final 72 °C step for 5 min. The JelAP1 oligonucleotide anneals to the adaptor and results in extension products directed towards the isolated genomic DNA fragment. The ISAba13_out and insH3_out oligonucleotides anneal close to the 3' end of their cognate insertion sequence. When used with JelAP1, they result in extension products across the insertion boundary with chromosomal DNA, towards the site of adaptor attachment. Thus, all products of the PCR reaction capture boundaries between ISAba13 and chromosomal DNA at insertion sites. Following PCR, each set of 8 equivalent reactions was pooled and products separated on a 2% (w/v) agarose gel. Products in the region of 400-650 bp were excised in a gel slice that was then stored at −20 °C for 30 min. The DNA was then recovered using a MinElute gel extraction kit (Qiagen) after 4 PE buffer washes.

In a second round of PCR reactions, we used an oligonucleotide (one of the 'Ap_P7_tagged' variants) that anneals to the same sequence as JelAP1 sequence and introduces a DNA barcode. This was in conjunction with a mixture of the 4 'staggered' oligonucleotides, all annealing to the same site towards the 3' end of the ISAba13 sequence, and containing different stagger lengths, to improve complexity of the final library. Resulting PCR products should contain the final 78+ bp (depending on the stagger length and insertion element) of the insertion sequence 3' end. The PCR conditions were the same as those used in step one above, except that the cycle number was reduced to 8. After PCR, DNA was recovered using a 1:0.75 ratio of AMPure XP beads. The size and purity of DNA fragments were assessed using a TapeStation, and concentration assessed by qPCR, before sequencing with an Illumina NovaSeq 6000 instrument. A schematic of the procedure is shown in Fig. S8. The raw data are available from ArrayExpress using accession codes E-MTAB-13792 and E-MTAB-13793 for experiments with *A. baumannii* and *E. coli* respectively.

## Native Tn-seq: bioinformatics

The FASTQ files were parsed using Barcode Splitter[85,86] (Galaxy version 1.0.1) to identify reads having the final 83 nt of ISAba13 at the 3' end. To account for differences introduced by the staggered oligonucleotides we allowed up to 4 mismatches and 2 deletions. Once selected, the first 83 nt of each read was removed using dada2:filterAndTrim[87] (Galaxy Version 1.20). Only remaining reads 20 or more nt in length were retained. These were then mapped to the reference genome (GenBank accessions CP144564, CP144560, CP144562, CP008707.1 for *A. baumannii* and U00096.2 for *E. coli*) using Bowtie 2[80] (Galaxy Version 2.4.2). Read depths for each DNA strand were then calculated using the BAM to Wiggle function of the RSeQC package[88] (Galaxy Version 5.0.1). Subsequent analysis of wiggle files, to identify sites of transposition, was done using logic functions in Microsoft Excel. First, we identified chromosomal locations where reads were detected but no reads mapped to at least 2 positions upstream. These were recorded as sites of ISAba13 insertion unless the read depth 1 position downstream was greater by 2-fold or more. In such cases, the latter position was used. Insertion sites identified in each biological replicate were then combined. Note that we did not expect, nor did we detect, the same sites of transposition in both replicates. Rather, we observed similar patterns of transposition (e.g. a bias towards H-NS bound DNA). When generating chromosome-wide heatmaps for insertions, we counted each position of insertion once, regardless of the number of sequencing reads corresponding to that position.

## ChIP-seq: library preparation and sequencing

These experiments were done as previously described[89]. Where needed, cells were transformed with a plasmid encoding H-NS-39. Cells were grown to mid-log phase, crosslinked using formaldehyde, and sheared nucleoprotein prepared exactly as in our prior work[89]. After immunoprecipitation, using 2 μl of anti-FLAG (Sigma-Aldrich, catalogue number F7425-2MG) in a 700 μl reaction, protein-DNA complexes

were washed to remove non-specific interactions before preparing libraries for Illumina sequencing. The raw data are assigned accession code E-MTAB-13790 and are available from ArrayExpress. Experiments were done in duplicate.

## ChIP-seq bioinformatics

Sequencing reads were aligned to the reference genome (GenBank accessions CP144564, CP144560, CP144562, CP008707.1) using Bowtie 2 as described above and coverage determined using the multi-BamSummary (Galaxy Version 3.5.4) function of deepTools2[90]. The data were visualised either as heatmaps, showing averaged read depth per 1 kb window, or in the Artemis genome browser[91], showing read depths averaged between replicates. The raw read depths are provided in Supplementary Data 1. To identify H-NS bound regions for the analyses in Fig. S2 and Table S2, we remapped the reads against the AB5075-UW reference chromosome (GenBank accession CP008706.1) so that co-ordinates matched existing essential gene lists[50]. We then determined the average H-NS ChIP-seq read depth in 50 bp windows. If the average of all windows within a given gene was >99 this was selected as an H-NS bound region. The cutoff was selected after visual comparison of H-NS bound and unbound regions.

## 3C-seq: library preparation and sequencing

Our procedure was based on the protocol of Rashid et al.[92]. Nuclease-free water was used throughout. Two biological replicates were done for each strain background and data shown are a combination of the contacts generated in each replicate. We began with a 10 ml culture of *A. baumannii* grown to an $OD_{600}$ of 0.5. Cells were recovered by centrifugation and resuspended in 4 ml of PBS before addition of 16 ml methanol. The samples were incubated on a roller bench at 30 rpm for 10 min at 4 °C. After recovery by centrifugation, cells were washed with 20 ml of ice-cold PBS and resuspended in 40 ml of 3% (*v/v*) formaldehyde diluted in PBS. The mixtures were again placed on the roller bench at 4 °C and left for 1 h. Glycine (400 mM) was used to quench the reaction during a further 15 min roller bench incubation. Cells were then recovered, resuspended in 5 ml of TE and divided into 1 ml aliquots, pelleted and flash frozen using liquid $N_2$. For cell lysis, we resuspended a cell aliquot in 50 μl of TE containing 0.5 μl of diluted lysozyme (made of up 0.5 μl Ready-Lyse lysozyme, 4 μl TE and 0.5 μl 1 M NaCl). The resuspension was then incubated at 37 °C for 15 min. The lysed cells were diluted to a final volume of 54 μl with TE before being mixed with 136 μl of water, 25 μl 10% (*w/v*) Triton X-100 and 25 μl 10x rCutSmart restriction digest buffer. We then added 10 μl of NlaIII (10 U/μl) for 3 h at 37 °C. Reactions were stopped by adding 12.5 μl of 10% (*w/v*) SDS and incubating for a further 20 min. Digested nucleoprotein was centrifuged for 1 h, at 4 °C and 25,000 × *g*. The DNA was then pelleted, resuspended in 200 μl water, and its concentration determined. Between 1-3 μg of digested nucleoprotein was diluted to 895 μl using water and 100 μl of 10x T4 DNA ligase buffer was added alongside 5 μl of 20 mg/ml BSA and 2 μl T4 ligase (2000 U/μl). Multiple reactions were set up to use all extracted nucleoprotein and incubated at 16 °C for 16 h. The incubation was then transferred to a Thermomixer for 1 h at 25 °C with shaking at 450 rpm. The reaction was stopped with 20.5 μl of 0.5 M EDTA. After ligation, RNA was removed with 16.6 μl of 10 mg/ml RNAse A for 30 min at 37 °C with shaking at 450 rpm. We then added 12.5 μl of 10 mg/ml proteinase K, and 120.8 μl of 5 M NaCl, for 16 h at 65 °C with shaking at 450 rpm. Samples were then pooled in a 15 ml Falcon tube before addition of 1 volume of 25:24:1 phenol:chloroform:isoamyl alcohol followed by vigorous mixing. The aqueous and organic layers were separated by centrifugation at 13,000 × *g* for 10 min at 4 °C. The upper aqueous layer was transferred to a new Falcon tube and the process was repeated. Next, 0.1 volume of 1 M sodium acetate pH 8.0, 0.025 volumes and 5 mg/ml glycogen, and 2.5 volumes of ice-cold ethanol were added before incubation at −20 °C overnight. The DNA was recovered by centrifugation at 25,000 × *g* and 4 °C for 20 min and the pellet washed with 0.5 ml ice-cold 70% (*v/v*) ethanol. The DNA was again pelleted and then dried in a 50 °C incubator. Each dried pellet was dissolved in 31 μl of water in a Thermomixer at 60 °C at 1000 rpm. After checking ~6 μl of the library using agarose gel electrophoresis, the remaining sample (which should contain at least 1 μg of DNA) was mixed with 60 μl of NEBNext sample purification beads in a low retention 1.5 ml microfuge tube. After mixing and incubation for 5 min at 25 °C beads were washed twice with 200 μl of 70% (*v/v*) ethanol. The DNA was then eluted with 25 μl of TE and the concentration determined. A minimum of 100 ng was then used to generate the final library using the NEBNext Ultra II FS DNA PCR-free library prep kit (high-input protocol). Libraries were sequenced using an Illumina NextSeq2000 instrument using a P1 600 cycle cartridge to generate paired end reads. The raw data have been deposited in ArrayExpress with the accession code E-MTAB-13800.

## 3C-seq: bioinformatics

The data were processed using the HiCUP pipeline described by Wingett et al.[93]. Briefly, sequence reads from 3C-seq experiments should contain 'di-tags' that are discontinuous parts of genomic sequence ligated via *Nla*III restriction sites. We used HiCUP truncator to detect *Nla*III restriction sites within individual sequence reads, and split the read at this sequence, prior to mapping. Hence, each half of the di-tag can be mapped to the correct genomic locus. Prior to mapping, HiCUP Filter was used to remove unwanted sequence reads. These include reads indicative of undigested (or religated) genomic DNA, self-ligated (i.e. circularised) sequences, or reads containing no *Nla*III site. Instances where the di-tag appears too large, based on input fragment size, were also excluded; these can indicate ligation of multiple restriction fragments. Any PCR duplicates were removed using HiCUP deduplicator. Finally, HiCUP digestor creates a set of *Nla*III fragments from the reference genome (CP144564, CP144560, CP144562, CP008707.1) and HiCUP mapper determines which restriction fragments align with each part of each di-tag. The resulting BAM file was processed with HiCUP to juicer converter (Galaxy Version 0.9.2). This generates a text file, containing pairs of genomic sites, conjoined at *Nla*III motifs, in legitimate di-tags. Lists of chromosomal contacts were further processed in OriginLab to generate contact matrices. The 10 kb and 1 kb contact matrix values are in Supplementary Data 2. Explorable versions of each matrix, aligned with H-NS ChIP-seq binding data, are provided in Supplementary Data files 3 and 4. When dividing one matrix by another, values in each matrix were first normalised by dividing the value in each bin by the total number of contacts in the matrix.

## Proteins

The *A. baumannii hns* sequence was cloned in plasmid pJ414 and the construct was used to transform *E. coli* T7 Express cells. Transformants were grown to mid-log phase before induction of H-NS expression with 1 mM IPTG for 2 h. Cells were collected by centrifugation and resuspended in 30 ml of 20 mM Tris-HCl pH7.2, 1 mM EDTA, 100 mM NaCl and 10% (*v/v*) glycerol with 1 cOmplete Mini EDTA-free protease inhibitor cocktail tablet. Cells were lysed by sonication and lysates cleared by centrifugation. Supernatant was filtered (0.4 μm pore size) and loaded onto a pre-equilibrated HiTrap heparin HP column. The protein was eluted with a gradient to 1 M NaCl. Fractions containing H-NS were identified by SDS-PAGE, pooled and dialysed overnight against 2 L of 20 mM Tris-HCl pH 7.2, 1 mM EDTA and 10% (*v/v*) glycerol. The resulting sample was loaded onto, and eluted from, a HiTrap SP XL column as described above. H-NS was recovered from the flow through and concentrated with a Vivaspin 20 MWCO 5000 column. The purified protein was then dialysed into 20 mM Tris-HCl pH 7.2, 300 mM KCl and 50% (*v/v*) glycerol for storage at −20 °C. The H-NS-39 peptide (N- MKKDQAIE-DAYNQIIEIAENVGFSVEQLLEFGAQKRKKTT-C) was synthesised and purified (Thermofisher) using Fmoc solid-phase technology.

## DNA bridging assays

The assay is based on the procedure of van der Valk et al. [56]. For each reaction, 3 μl of MyOne streptavidin T1 Dynabeads (Thermofisher) were added to a microfuge tube, washed once with 50 μl PBS and twice with 50 μl of 20 mM Tris-HCl pH 7.9, 2 mM EDTA, 2 M NaCl, 2 mg/ml Acetylated BSA, 0.04% (v/v) Tween-20. The beads were then resuspended in 3 μl of the latter buffer. One hundred fmol of biotinylated bait DNA, in 3 μl of 10 mM Tris-HCl pH 7.9, 50 mM KCl and 10 mM MgCl$_2$ was added to the resuspended beads and incubated at 25 °C for 20 min with shaking at 1000 rpm. Beads were washed twice with, and then resuspended in, 16 μl of 10 mM Tris-HCl pH 7.9, 0.02% (v/v) Tween-20, 1 mg/ml Acetylated BSA, 1.25 mM Spermidine, 12.5 mM MgCl$_2$, 6.25% (v/v) glycerol, 1.25 mM DTT. Proteins and radiolabelled prey DNA were then added and samples incubated at room temperature for 20 min. Beads were washed twice with 20 μl of reaction buffer and resuspended in 12 μl of 10 mM Tris-HCl pH 7.9, 1 mM EDTA, 200 mM NaCl and 0.2% (w/v) SDS. A 5 μl aliquot of each reaction was spotted on a nitrocellulose filter paper and exposed to a Fuji phosphor screen overnight. After imaging with an Amersham Typhoon instrument, spot intensity was quantified using ImageJ software.

## Electrophoretic mobility shift assays

Experiments were done according to the protocol of Haycocks et al. [89]. Briefly, radiolabelled DNA fragments were mixed with herring sperm DNA (12.5 mg/ml) and indicated proteins in 40 mM Tris-acetate pH 7.9, 10 mM MgCl$_2$ and 100 mM KCl. After incubation at 37 °C for 10 min, samples were analysed by electrophoresis using a 5% (w/v) polyacrylamide gel. After drying, gels were exposed to a Fuji phosphor screen overnight and imaged using an Amersham Typhoon.

## Reporting summary

Further information on research design is available in the Nature Portfolio Reporting Summary linked to this article.

## Data availability

All Illumina sequencing data generated in this study have been deposited in ArrayExpress under accession codes: E-MTAB-13791, E-MTAB-13792, E-MTAB-13793, E-MTAB-13790 and E-MTAB-13800. Genome sequences are available from GenBank with accession codes: CP144559, CP144563, CP144560, CP144562, CP144564 and CP144565. Results generated from processing of Illumina sequencing data (e.g. differential gene expression analysis) are available as Source Data or Supplementary Data files. Original gel images are provided as Source Data. Source data are provided with this paper.

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

## Acknowledgements

We are indebted to Keith Derbyshire, Joseph Wade, Ronald Chalmers, Stephan Uphoff, Clémence Whiteway and Charles Van der Henst for helpful discussions. R.L.W. acknowledges support from Genomics Birmingham. This work was funded by a Wellcome Trust Investigator award (no. 212193/Z/18/Z) to D.C.G. and an AAMR doctoral training programme studentship to C.C.

## Author contributions

All experimental work was done by C.C. except for ChIP-seq (done by S.L.) and 3C-seq (R.L.W. and J.R.J.H.). The RNA-seq and DNA bridging experiments were done with support from D.F. and P.S. respectively. Experiments were conceived by D.C.G. and C.C. and all authors contributed to data analysis. The manuscript was written by D.C.G. with all authors providing comments.

## Competing interests

The authors declare no competing interests.
