## [Peer Review File · Nature Communications]

H-NS is a bacterial transposon capture proteinReviewer #1 (Remarks to the Author):

Reviewer comments

This strong, impactful work describes how a nucleoid structuring protein, H-NS (present in many important bacterial pathogens) acts as a transposon capture protein. This is a newly characterized activity for H-NS, which is well-known to modulate DNA structure on a local scale and silence transcription of genes in its vicinity. The work exploits a variant of standard Tn-seq, called native Tn-seq (where DNA amplification steps are minimized to avoid bias generated by the native IS elements being studied) allowing native transposition events to be mapped. While most of the study is done in the clinically relevant bacterium *Acinetobacter baumannii*, in the latter stages, the authors re-examine some *E. coli* data to see if their observations hold true for this bacterium. They do, which broadens the findings to other bacteria nicely. In *Acinetobacter baumannii*, this phenomenon drives IS element capture in regions bound by H-NS, a common feature of pathogenicity islands or loci involved in virulence. Pathogens' genetic plasticity is of broad interest. Given the association of H-NS with virulence gene loci in other pathogens, it seems likely that H-NS may drive transposon capture at these medically-relevant, genetic loci too.

The paper flows logically, is written well and the techniques are innovative. Additional clarifications statistical analyses, and further thoughts about Figure 1 and its placement, would improve the manuscript. Major and minor comments are found below.

Major Comments

1) H-NS 39 – Clarification about H-NS 39 is required. It is clear that this is a 39 amino acid peptide sourced from H-NS, but it is not clear which 39 amino acids are represented. Is it the first 39 amino acids (from the dimerization domain) or 39 amino acids taken from the dimer-dimer interface, described by Van der Valk et al., 2017 as HNS 56-82 (Importantly only 26 amino acids – hence the confusion)? The latter seems more likely, but nowhere do the authors describe this peptide, or allele that makes it, clearly. Additionally, there is no description of who purified this peptide – was it ThermoFisher or you? L558 is written ambiguously- clearer details are needed. In a similar vein, what were the growth conditions of cells that were expressing this allele from the inducible plasmid? What is the copy number of this plasmid, and how was the allele induced? This is relevant for every *in vivo* assay where H-NS 39 was used – there are many instances. In your *in vivo* experiments, what is the ratio of H-NS to HNS 39 and how it compares to the ratio used *in vitro* experiments (12:1 through 3:5 for the bridging assay & 6:1 to 3:5 in EMSAs)? These details matter because if the HNS 39 allele expresses amino acids 1-39 (question asked above), it is well established that high concentrations of this allele can interfere with the binding of H-NS and other H-NS paralogs, like StpA – causing a dominant negative effect (Picker et al., 2016 doi: 10.3390/genes7120112). Of note, *A. baumannii* does have an *stpA* gene, so this concern is relevant. Why is StpA never mentioned (see future comment).

2) Throughout this study, statistical tests are rarely used. A full list of places where statistical analyses are required is provided below:

- Fig. 1d: While the difference between cell variants is striking, statistical analyses are absent. A two-tailed student's T-test is required.
- Fig. 1e: Statistics between conditions are not present. Because the represented data is the average of three independent experiments a one-way ANOVA with posthoc Bonferroni can be used for each given time point. Additionally, at what time point does the most drastic growth difference occur?
- Fig S1b-d: Statistical analyses are absent between cell variants. A two-tailed student's T-test is recommended.
- Fig. 2d: Whiskers represent the 5th -95th percentile (stated L824, fig legend) however, traditionally these are used for the min and max values. Why are the whiskers represented in this manner? And why is 70-80% AT content the only dataset with a full box and whisker plot? This seems worthy of comment in the figure legend.
- Fig. 3c: Statistical analyses are absent between strains and no error bars are included. The lack of stats suggests n may equal 1? How many representative trials were done? If the graph represents a summation of total insertions then these data might be better displayed in a Table. A two-tailed student T-test is required if n>1.

- Fig. 3d: Again, whiskers represent the 5th - 95th percentile. Traditionally these are used for the min and max values. Why are the 5-95 percentiles being shown whiskers represented in this manner? This seems worthy of comment in the figure legend.
- Fig. 4a: Statistical analysis of this figure is absent. Using an ANOVA with the appropriate posthoc test (Tukey-HSD) would strengthen the argument that including H-NS-39 leads to a significant decrease in DNA recovery.
- Fig. S2b: Statistics are included, however, the p-value used is not included (what confidence interval was used? The correct way to write p-values is lowercase and italicized. Furthermore, the better statistical analysis to perform would be a two-way ANOVA with posthoc Tukey HSD. Then, to compare between wt and hns-, independent sample t-tests can be used to assess statistical significance between the means of the two groups. There are no error bars – please add them.

3) Figure 1 – This figure attempts to set the stage for the paper, but multiple panels have issues that will cause readers, especially those with knowledge of bacterial pathogenesis, to get hung up on technicalities, accuracy, statistics, missing genes and nomenclature. Would the relocation of this figure to the end of the manuscript be better? It would better reflect the abstract and there would be fewer distractors before getting to the meat of the paper. A list of issues with Figure 1 is presented below: -

- Fig. 1a – the grey colonies are likely mucoid colonies – mucoid and non-mucoid phenotypes are observed in Rakovitsky et al., 2021 (doi: 10.1093/ofid/ofab386), Hu et al., 2020 (doi: 10.2147/IDR.S230178), and reviewed in Shan et al., 2021 (doi: 10.1016/j.micres.2022.127057). Has a string test been done? You could do this easily (Kon et al., 2020 DOI: 10.1186/s12866-020-01971-9). I find the failure to better characterize these grey colonies underwhelming.
- Fig.1b - There are discrepancies in the genomic representation of the K-locus. The authors cite Kenyon & Hall which illustrates the K-locus however, they do not include the AB5075 strain. Upon inspection, Senchenkova et al., 2015 (doi:10.1016/j.carres.2015.02.011) describe the K-locus of the AB5075 strain that the authors depict, however the loci do not match. In the present work, *itrA1* is absent between *gtr52* and *qhbA/weeH*. This work should be cited. Kenyon & Hall describe the *qhbA* gene being homologous to *weeJ* however, the authors label this gene as *weeH*. This is confusing. Also, how many promoters are active in this locus? Are we looking at an operon? This needs clarification.
- Fig 1c and Table S1 - Why are only some differentially expressed genes highlighted? Was a threshold level of change used? If so, what was it? I assume the title in Table S1 should be Log₂ fold change? Please fix this. Was Log₂ fold change the cut-off? If so, please state clearly. Given results in panels f-h, other genes involved in motility or biofilm formation are likely to be differentially expressed egs. *BfmRS*, which regulates EPS and pili production, *OmpR/EnvZ* bacterial envelope stress – but these are not seen in the figure or the Table – why not? Are these genes encoded by the large plasmid (83 kb) that is not stably maintained (see whole genome sequencing section L350? In this regard, complementing the mutant with the ISAbA13 with an exogenous copy of the locus would help determine which phenotypes are caused by the IS element and which are artifacts of large plasmid loss. This part of this study is not methodical.
- Fig. 1e: The growth defect exhibited by the grey mutant appears restored after 24 hours. Has this strain reverted to the wild-type phenotype or obtained a second site suppressor? What is happening here? If the IS element has jumped again this would be relevant to include.

Minor Comments

Abstract L38-40 “For example...” is an odd way to start a sentence

L48 There is a comma missing before traits. Add hyphen “horizontally-acquired”

L57 Add Picker et al., 2016 doi: 10.3390/genes7120112 to support that H-NS is conserved throughout the gamma Proteobacteria

L88 What is the position of the K-locus in AB5075 strain? The position of the two other ISAbA13 is provided. Add the position at the K-locus too.

L94 Change tex. 45-fold effect is log₂ of -5.94

L133 What is the nature of the hns mutant? Table S4 says T26 transposon in hns, but details are missing. Where is the transposon? Does this generate a true null, or is some part of the HNS protein made? This is important as N terminal truncations of HNS display dominant negative effects on other HNS paralogs and *Acinetobacter* has a *stpA* gene. Thus, there is scope for an HNS

truncation to interfere with StpA binding. Please address the nature of the hns mutant that you've used.

A comment about this section generally – What effect does temperature or osmolarity have on transposon capture? If H-NS is more tightly bound at 30°C wouldn't you expect more capture at these ambient temps than at host temps? It would be odd for a pathogen to exhibit heightened mutability outside of its host – normally we consider pathogens to become more genetically variable in the host environment. Is StpA, the well-established H-NS paralog, involved in IS element capture? If the hns mutant is a true null then the answer is no, based on your data, but the lack of details makes it challenging to discern, emphasizing the need to include the nature of the hns mutant used.

L145 "Two plasmids... contain regions of elevated transposition"- which two? Be specific. If it is the two largest ones is this due to the amount of DNA being higher i.e. larger plasmid, more DNA, more likely to be hit. Please clarify this point.

L165 It is unclear how IS elements that hit the essential genes are being counted? Wouldn't these be eliminated as these mutations would be lethal? Are you picking up signatures from dead bacteria? Or are the insertions in positions within the essential gene that hobble the gene without knocking it out completely? Some further explanation is warranted.

L197 What is the purpose of the statement about the weaker secondary diagonal? Is it weaker? That is not abundantly clear. This is never mentioned again – it is almost an orphaned thought. Furthermore, doesn't the sentence in L199 contradict this statement? This causes confusion.

L205 It would be good to tie this observation back to Vicky Lioy's work in *E. coli*, where H-NS makes short-range interactions (Liyo et al., 2018; <https://doi.org/10.1016/j.cell.2017.12.027>)

L210 In this section and throughout the paper it is unclear how the HNS39 allele was expressed in vivo – this is essential information and must be included.

L211 Change the text to "Our observations are consistent with HNS 39 impeding the ability of native H-NS to bridge, under the conditions used or under the expression levels used in vivo. This would be more accurate and relates to concerns about ratios of HNS and HNS 39 raised in the major comment #1.

Section starting L221 – This section would benefit from a more concrete concluding statement that explains which of the two scenarios pertaining to pre- post excision, raised on L 223-225, is more likely based on your data.

L260 Is IS903 an IS5 derivative? What about InSH3 (L271)? This is worthy of comment, given paragraph 1 of the Discussion.

Discussion points to consider for inclusion –

- i) H-NS increases *Acinetobacter baumannii* competence – Le et al., 2021 this is relevant in paragraph #2 where you discuss horizontal gene transfer and H-NS.
- ii) Other NAPS – are they involved in this type of process? Is there any evidence?
- iii) What is the potential for involvement of StpA in this phenomenon?
- iv) Environmental parameters that affect H-NS DNA engagement, are likely to affect IS element capture too, right? How does this relate to pathogen genetic plasticity – there's scope to discuss this.

Methods

L366 while Whiteway is cited, the original methods are described in Kon et al 2020 (DOI: 10.1186/s12866-020-01971-9) Replace or add this citation.

L376 The procedure was first described by Tomarus et al 2008 and Allen 2020. Add or replace these citations.

L384 Please include how the cells were grown immediately prior to RNA-seq library preparation. These details are important.

References

L691 – Van der Valk citation #50 is incomplete. This is a repeating problem in this section.

Figures

Figure 1 – see major comment #3

Figure 3b What are the genetic loci shown? Add locations or some descriptor.

Figure S2a What are the genetic loci shown? Add locations or some descriptor.

Reviewer #2 (Remarks to the Author):

Reviewer #3 (Remarks to the Author):

The manuscript by Cooper et al. reveals a new role of H-NS in regulating transposition. Unexpectedly, the authors show that, rather than insulating the DNA from transposable elements, H-NS targets transposition to DNA regions that it is known to silence, such as virulence genes and prophages. This study uses mainly NGS-based technologies to investigate H-NS effects on transposition of ISAbA13 element in *A. baumannii*, which can cause serious infections. The results demonstrate that H-NS can promote the insertion of transposons by physically capturing them and that the capture is mediated by DNA bridging activity. The authors also provide explanations of apparent discrepancies between their data and those published previously (mostly with the *E. coli* H-NS) and data analysis consistent with a notion that *E. coli* H-NS favors transposition into genomic loci it silences.

The presented experimental data are convincing, the manuscript is well written, and the logic is clear. Overall, these findings are significant for our understanding of bacterial evolution and the role of NAPs therein and would be of interest to broad audience. However, this reviewer has a number of suggestions that may help to make the manuscript more accessible.

Major:

1) A major conclusion is that H-NS manipulates transposition to maximise favorable evolutionary outcomes for the host, by preferentially targeting transposons (at least this one) into foreign DNA that is silenced by H-NS. I think that a separate emphasis on plasmids and prophages, which are ultimate foreign DNA, is not warranted. While, as stated in line 302 "by facilitating transposition into prophage or plasmids, H-NS provides a route via which transposable elements can spread horizontally", how many H-NS binding sites are located in prophage/plasmids? If the binding sites in prophage/plasmid are not the majority, the authors cannot single out prophage or plasmids as "special" transposition targets. This information should be discussed using the ChIP-seq data in the manuscript.

2) Any explanation for the high number of insertions in the H-NS unbound region? Figure S2a shows a high peak of insertion sites in essential genes of wt cells. What is the essential gene?

3) H-NS-39, a truncated variant of H-NS, has been used to argue that the DNA bridging activity of H-NS is responsible for transposon capture. However, the *in vivo* influences of H-NS-39 on DNA bridging are limited. How many sites are influenced? Figure 4 and S4 show only 5 sites, and all these sites have low H-NS occupancy. An assumption here is H-NS-39 abolishes DNA bridging. What if H-NS binds to the transposase to increase transposition and 39 peptide blocks this interaction?

4) The argument for "Transposition events are likely captured after ISAbA13 excision" needs to be elaborated. For example, line 237 states "This contrasts sharply with transposition frequency", but the meaning of this observation is not very clear. How to make sense of "proximity to an existing copy of ISAbA13 could impact transposition independently of H-NS" since the two indicated regions (especially ISAbA13 copy 1 area) do not have significantly higher peaks than others in Figure S5c. With that being said, consider moving Figure S5 to the main figures.

Minor:

- 1) Are the positions 1.764 Mb and 3.863 Mb covered by H-NS? How does the ISAb13 dissociate from H-NS after capturing? Any thoughts on this?
- 2) Does *A. baumannii* have another homolog of H-NS? *E. coli* has H-NS and StpA.
- 3) Line 70: reference 30 is not the latest list. Please use the WHO bacterial priority pathogens list, 2024 (<https://www.who.int/publications/i/item/9789240093461>).
- 4) Fig. 1a and Fig. S1a are basically the same. If anything, Fig. S1a can show more information.
- 5) Line 93: "ISAb13 appears upregulated". What is the read count for ISAb13? There is no information in the Method describing the read-count cut-off for differential gene expression analysis.
- 6) Line 97: "mediated by a type IV pilus requiring pilA". Citation?
- 7) Line 123: Fig. 2b cannot clearly show that "transposition is biased towards non-coding sequences". Another pie chart that shows what proportion of the genome each class occupies needs to be shown, or % genome can be added inside the slices..
- 8) Fig. 2c: How is the correlation coefficient of Fig. 2c calculated? What is the gene region of the expansion?
- 9) Line 150: should it be "targeting transposable elements in prophage and plasmids"?
- 10) Line 208: define hairpin loop
- 11) Line 328: "Lysogeny Broth".
- 12) Line 331: Italicize "*A. baumannii*".
- 13) Lines 355-356: please cite publications for these bioinformatic tools. Also, how are the reads trimmed? The reviewer recommends the authors check out "minimap2" for nanopore reads alignment in the future as minimap2 has replaced bwa-mem for long-reads alignment (<https://github.com/lh3/bwa>).
- 14) Lines 361-364: please consider using the name p1AB5075 for this 83.61 kb plasmid. If the p1AB5075 was lost, why can it be assembled from Illumina sequencing data? If only part of the population lost p1AB5075, the authors should make efforts to get more sequencing depth with nanopore and assemble this plasmid. The assembled plasmids should be included in the NCBI Genome Assembly submissions (GCA_036602705.1 and GCA_036601155.1) to make a true "Complete Genome" assembly level. The p1AB5075 has antibiotic-resistance genes, which can be used for testing its presence.
- 15) Line 369: an extra period mark.
- 16) Line 386: "as described previously", insert citation?
- 17) How many repeats for RNA-seq and ChIP-seq?
- 18) Lines 398 and 483: 50 bp windows and 99 was selected as the threshold. If the authors had a good reason for the cutoffs, please explain it in the Method. The readers will benefit from it.
- 19) Lines 406 and 408: "dH2O" is not a common word for everyone to understand. Please use the full name.
- 20) Line 410: "5 ul 0.5 M EDTA" doesn't make sense because no total reaction volume is provided.
- 21) Are the coverages for detected sites of transposition reasonable? Are they all higher than 30?
- 22) Line 459: "read depth 1 position"?
- 23) Line 544: "dividing the value in each bin my the"?
- 24) Lines 552, 553: editing "heparin HP1", "against)".
- 25) Line 569: what is the reaction buffer?
- 26) Line 792: and grey variant cells.
- 27) Line 809: depth
- 28) Which cells are used for native Tn-seq? Wild-type or the grey variant. If wild-type, please show the depth of 1.764 Mb and 3.863 Mb locations for comparison.
- 29) Figure S2b, how were insertions per kb calculated? Is the method different from Fig. 3d?
- 30) Figure 4a, the surface for DNA binding is also dark green. Is the dimerization surface pale green?
- 31) Figure 4d can be moved to supplement. Fig. 4e, it's not easy to see the difference. Please indicate the actual values for the discussed part. The same goes for Figure S4.
- 32) Figure S4b legend: "b. a. Changes"?
- 33) Why are the correlation coefficients in Figure 5a and Figure 3a different? Maybe use "correlation coefficients (r)" in the legend.
- 34) Figure 5c, is this wt or H-NS deletion cell?
- 35) Figure S6a, what is the correlation coefficient?

Reviewer #4 (Remarks to the Author):

The paper by Cooper and colleagues focuses on a largely neglected function of the nucleoid-associated protein H-NS in bacteria: its ability to influence target site selection by transposable elements. H-NS, a major nucleoid structuring protein in gamma proteobacteria, binds to hundreds of sites in the bacterial chromosome, silencing the expression of genes relevant for host invasion or environmental adaptation. This paper shows that sites bound by H-NS constitute hotspots for the insertion of ISAbal3, an IS5-type transposon in *Acinetobacter baumannii*. Interestingly, the targeting of H-NS-bound sites by the transposon is mediated by H-NS's DNA-bridging activity; a finding that lays the ground for future studies of the mechanism involved. The paper also discusses preliminary evidence that suggests a similar trend in *E. coli*, pointing to the generality of the phenomenon.

While the participation of H-NS in transposition was recognized prior to this work, its role in target capture has never been documented with the genome-wide depth achieved in this study. Given the relevance of H-NS-controlled processes in *A. baumannii* pathogenicity, the phenotypic diversity resulting from the insertional disruption of H-NS-regulated genes is an important aspect of the biology of this medically significant bacterium. The data in the paper are clearly described and convincing overall. The clinical implications of the work can be expected to attract the attention of a wide readership. I only have a couple of comments that the authors may want to consider in revising the manuscript.

1. An obvious shortcoming of the "Native Tn-seq" procedure used in this study is the potential bias in the profile of transposon insertion sites that might result from differential growth rates of the insertion mutants. The method description (line 400) indicates that genomic DNA was extracted from a mid-exponential phase culture, sub-cultured from an overnight culture (lines 402-403) presumably inoculated from a single colony. It is highly likely that most insertion mutants identified in the subsequent analysis were already present in the starting colony (which typically contains approx. 10^9 cells). Their undergoing several divisions prior to cell harvesting may significantly affect their relative representation. The authors should discuss this point and provide more details on how these steps were performed: were the biological replicates inoculated from separate single colonies? What fold-dilution was used to sub-culture the overnight culture? In the methods section describing strain handling (line 326), the authors state: "To avoid the use of sub-cultures, inoculated liquid media was left at room temperature overnight before being transferred to 37°C, with shaking, until cells reached the required growth phase" (lines 328-330). For which experiment was this method used? Although eliminating the sub-culturing step is a good idea, adding a room-temperature static growth step may introduce additional bias; for instance, mutants adhering to the tube walls may be left behind. Why not perform the entire culture expansion from colony to the "required growth phase" in one step, with shaking, at 37°C? This would minimise the number of cell divisions without introducing unnecessary changes in growth conditions. Note, however, that the most rigorous way to perform this analysis, which the authors might consider for future work, would be to engineer a strain carrying an inducible transposon and prepare the DNA library shortly after induction.

2. The authors may want to elaborate on their vision of how DNA-bridging by H-NS could enhance target capture by the transposon. Plausible scenarios can be inferred from the notion that a crucial step in target capture involves transposase-mediated bending of the DNA. It is conceivable that factors promoting bending at specific sites might facilitate transposon insertion at these locations. Could the apexes of DNA loops or plectonemes stabilized by H-NS-mediated bridging provide such favorable bends?

Minor points.

3. Line 85. Add "a" to Fig. S1.

4. Line 116. Not clear what bias the authors are referring to.

5. Line 121. Whiteway et al; insert reference number.

6. Line 188. Insert "the" between "understand" and "effects".

7. Line 312. Strictly speaking the term "scarless" applies to the reversibility of the ISAbal3

insertion, not to the insertion itself.

8. Lines 400-447. Native Tn-seq section. Providing a schematic diagram of the step involved in the library preparation as a supplementary figure would help understanding the workflow.

9. Line 468. "...chromosome was altered to encode H-NS with a C-terminal 3xFLAG fusion". This is a rather cryptic and unorthodox way to describe a strain construction. Please provide details.

10. Line 950. Remove "a" after "b".

11. Figure 1e. The filled circles plot is invisible. Consider splitting the diagram (placing active and inactivated serum plots in separate side-by-side panels).

12. Figures 3b, 5b, S2a, S3a right, S3b. Please add chromosome position of genes (Mb); add gene names in S2b and S3a right.

13. Figure 4d. What are those "empty" bins in the terminus region?

Reviewer #5 (Remarks to the Author):

We thank the reviewers for their supportive comments and helpful suggestions. It's also nice to see that early career researchers have had the opportunity to review our paper. We'd like to commend them one doing a good job. Our responses our outlined below.

REVIEWER COMMENTS

Reviewer #1 (Remarks to the Author):

Reviewer comments

This strong, impactful work describes how a nucleoid structuring protein, H-NS (present in many important bacterial pathogens) acts as a transposon capture protein. This is a newly characterized activity for H-NS, which is well-known to modulate DNA structure on a local scale and silence transcription of genes in its vicinity. The work exploits a variant of standard Tn-seq, called native Tn-seq (where DNA amplification steps are minimized to avoid bias generated by the native IS elements being studied) allowing native transposition events to be mapped. While most of the study is done in the clinically relevant bacterium *Acinetobacter baumannii*, in the latter stages, the authors re-examine some *E. coli* data to see if their observations hold true for this bacterium. They do, which broadens the findings to other bacteria nicely. In *Acinetobacter baumannii*, this phenomenon drives IS element capture in regions bound by H-NS, a common feature of pathogenicity islands or loci involved in virulence. Pathogens' genetic plasticity is of broad interest. Given the association of H-NS with virulence gene loci in other pathogens, it seems likely that H-NS may drive transposon capture at these medically-relevant, genetic loci too.

The paper flows logically, is written well and the techniques are innovative. Additional clarifications statistical analyses, and further thoughts about Figure 1 and its placement, would improve the manuscript. Major and minor comments are found below.

Major Comments

1) H-NS 39 – Clarification about H-NS 39 is required. It is clear that this is a 39 amino acid peptide sourced from H-NS, but it is not clear which 39 amino acids are represented. Is it the first 39 amino acids (from the dimerization domain) or 39 amino acids taken from the dimer-dimer interface, described by Van der Valk et al., 2017 as HNS 56-82 (Importantly only 26 amino acids – hence the confusion)? The latter seems more likely, but nowhere do the authors describe this peptide, or allele that makes it, clearly. Additionally, there is no description of who purified this peptide – was it ThermoFisher or you? L558 is written ambiguously- clearer details are needed.

We have added the sequence of the 39 aa peptide to the methods section and modified the text so it is clear ThermoFisher did the purification. We chose the exact position to start and stop the peptide based on AlphaFold predictions. We have added a new figure (Figure S4) showing an AlphaFold 3 prediction of the 39 aa peptide and its interaction with full length H-NS. The figure also shows the below alignment of *E. coli* and

A. baumannii H-NS, where the red sequences correspond to the peptide used in the van der Valk paper and H-NS-39.

```

coli          MSEALKILNNIRTTLRAQARECTLETLEEMLEKLEVVVNERREEESAAAAEVEERTTKLQQ
baumannii    -----MKPDISELSVEELKRLQEEAEALIASKKD-----QAIEDAYNQ
              :::: * ::* *::: * : *::: :::: :      :*

coli          YREMLIADGIDPNELLNSLAAVKSGTKAKRAQRPAYSYVDENGETKTWTGQGRTPAVIK
baumannii    IIIEIAENVGFSVEQLLEFGAQRK----KTTRKSVEPRYRNKNNAEETWTGRGKQPRWL
              *:      *:. :::*: * :. * :::: : * :*: :****:*: * :

coli          KAMDEQGKSLDDFLIKQ
baumannii    AEIEK-GAKLEDFLI--
              ::: * .*:****

```

In a similar vein, what were the growth conditions of cells that were expressing this allele from the inducible plasmid? What is the copy number of this plasmid, and how was the allele induced? This is relevant for every in vivo assay where H-NS 39 was used – there are many instances.

Details re growth conditions are in the first section of the methods. The text has been modified to make this clearer. The hns-39 allele was expressed from the native hns promoter in plasmid pVLR1Z (i.e. H-NS-39 is not over produced). Based on measurements by the authors who made this plasmid, it is likely to be present at around 50 copies per cell. These details have also been added to the methods section.

In your in vivo experiments, what is the ratio of H-NS to HNS 39 and how it compares to the ratio used in vitro experiments (12:1 through 3:5 for the bridging assay & 6:1 to 3:5 in EMSAs)?

It's difficult to answer the question exactly but we can infer that H-NS-39 will be in excess, based on the information above. Importantly, our experimental measurements are consistent with full length H-NS remaining DNA bound (confirmed by ChIP-seq) but bridging being disrupted (3C-seq).

These details matter because if the HNS 39 allele expresses amino acids 1-39 (question asked above), it is well established that high concentrations of this allele can interfere with the binding of H-NS and other H-NS paralogs, like StpA – causing a dominant negative effect (Picker et al., 2016 doi: 10.3390/genes7120112). Of note, A. baumannii does have an stpA gene, so this concern is relevant. Why is StpA never mentioned (see future comment).

We believe that the reviewer is mistaken; A. baumannii strain AB5075 does not encode StpA. We have rechecked this using the E. coli StpA amino acid sequence to search for a homologue in AB5075. Only two hits are returned. One is the chromosomal H-NS gene, studied in our paper, the other is a plasmid encoded H-NS-like protein. Importantly, these A. baumannii proteins are very similar to each other but neither closely resembles StpA (see alignments below).

```

AB5075_hns      MKPDISELSVEELKRLQEEAEALIASKKDQAIEDAYNQIIEIAENVGFSVEQLLEFGAQK
AB5075_plasmid_hns  -MSQISELSIEELKDLQLEAAKLIEIKEEQAIEDAYFKIISIAESVGYSVEDLLKVGAAAS
      .:*****:***** * * * * * * *:***** :*:***.*:***:***:*. * .

AB5075_hns      RKKKTRKSVPEPRYRNKNNAEETWTGRGKQPRWLVAEIEKGAKLEDFLI-
AB5075_plasmid_hns  SKKARKSVKPRYRSKANEQDTWTGRGKQPRWLVAEIEKGAKLEDLRIN
      ** .:*****:***** * * *:*****:*****:*****: *

AB5075_hns      -----MKPDISELSVEELKRLQEEAEALIASKKDQ-----AIEDAYNQ
Ecoli_stpA      MSVMLQSLNNIRTLRAMAREFSIDVLEEMLEKFRVVTKERREEEQOQRELAERQEKIST
      :.. *::: *::: * : .. : : : : : : : :

AB5075_hns      IIEIAENVGFSVEQLLEFG---AQKRKKTTRKSVPEPRYRNKNNAEETWTGRGKQPRWL
Ecoli_stpA      WLELMKADGINPEELLGNSSAAAPRAGKKRQPRPAKYKFTDVNGETKTWTGQGRTPKPIA
      :*: : *:. *:* * . : * * :..: : : * . :***:*: * : :.

AB5075_hns      AEIEKGAKLEDFLI
Ecoli_stpA      QALAEGKSLDDFLI
      : :* .*:****

AB5075_plasmid_hns  -----MSQISELSIEELKDLQLEAAKLIEIKEEQAIEDAYFK-----
Ecoli_stpA      MSVMLQSLNNIRTLRAMAREFSIDVLEEMLEKFRVVTKERREEEQOQRELAERQEKIST
      . : : * * : : : : : * : * * * * :

AB5075_plasmid_hns  IISIAESVGYSVEDLLKVGAAASS---KKKARKSVKPRYRSKANEQDTWTGRGKQPRWL
Ecoli_stpA      WLELMKADGINPEELLGNSSAAAPRAGKKRQPRPAKYKFTDVNGETKTWTGQGRTPKPIA
      :..: : * . *:* .*: : : * * :..* : : . * .***:*: * : :.

AB5075_plasmid_hns  AEIEKGAKLEDLRIN
Ecoli_stpA      QALAEGKSLDDFLI-
      : :* .*: * *

```

The plasmid encoded H-NS, which we call H-NS-2, is of course interesting, and we are currently working on it. We already know that H-NS-2 has no impact on global patterns of transposition (compare WT and $\Delta hns2$ below, unpublished data). We think that *hns2* may be having plasmid specific effects, but this is very much a preliminary observation. It is for this reason that we included the sentence “This may explain why phage and plasmids encode H-NS modulating factors” at the end of paragraph 2 in the Discussion.

2) Throughout this study, statistical tests are rarely used. A full list of places where statistical analyses are required is provided below:

Please see responses below.

- Fig. 1d: While the difference between cell variants is striking, statistical analyses are absent. A two-tailed student's T-test is required.

This has been done.

- Fig. 1e: Statistics between conditions are not present. Because the represented data is the average of three independent experiments a one-way ANOVA with posthoc Bonferroni can be used for each given time point. Additionally, at what time point does the most drastic growth difference occur?

This has been done and the P values are in the figure legend. By "most drastic" does the reviewer mean the time point where the difference between conditions returns the smallest p-value? If so, this is after 3 hours. If the reviewer means the largest difference in OD₆₀₀, this is after 11.5 hours.

- Fig S1b-d: Statistical analyses are absent between cell variants. A two-tailed student's T-test is recommended.

This has been done.

- Fig. 2d: Whiskers represent the 5 th -95 th percentile (stated L824, fig legend) however, traditionally these are used for the min and max values. Why are the whiskers represented in this manner? And why is 70-80% AT content the only dataset with a full box and whisker plot? This seems worthy of comment in the figure legend.

Apologies, we don't understand why the software wasn't showing the full plot for all categories. We have remade the plots and changed the whiskers to minimum and maximum values.

- Fig. 3c: Statistical analyses are absent between strains and no error bars are included. The lack of stats suggests n may equal 1? How many representative trials were done? If the graph represents a summation of total insertions then these data might be better displayed in a Table. A two-tailed student T-test is required if n>1.

These are total numbers of insertions detected across two biological replicates for each strain, hence error bars are not appropriate. We appreciate that this could also be presented as a table but would prefer to keep the bar chart. We have added the detail that these are total insertions, rather than averages, to the legend.

- Fig. 3d: Again, whiskers represent the 5 th - 95 th percentile. Traditionally these are used for the min and max values. Why are the 5-95 percentiles being shown whiskers represented in this manner? This seems worthy of comment in the figure legend.

Altered as for the equivalent example above

- Fig. 4a: Statistical analysis of this figure is absent. Using an ANOVA with the appropriate posthoc test (Tukey-HSD) would strengthen the argument that including H-NS-39 leads to a significant decrease in DNA recovery.

This has been done.

- Fig. S2b: Statistics are included, however, the p-value used is not included (what confidence interval was used?)

We do not understand what the reviewer means by “the p-value used is not included”.

The correct way to write p-values is lowercase and italicized.

To our knowledge, there is no standardised way to write “p” with both “P” and “*p*” being common. If the journal has a preferred house style then this will be applied during typesetting.

Furthermore, the better statistical analysis to perform would be a two-way ANOVA with posthoc Tukey HSD. Then, to compare between wt and hns-, independent sample t-tests can be used to assess statistical significance between the means of the two groups. There are no error bars – please add them.

In this situation, we prefer to keep the current analysis using t-tests to make pairwise comparisons between specific groups of genes (we note that the referee suggests using t-tests for the similar data in Figure 3c). We have not shown error bars because these are absolute measurements of insertions per kb for different gene groups (i.e. the error bars would not be reporting any sort of experimental variation, unlike other situations in which we show error bars).

3) Figure 1 – This figure attempts to set the stage for the paper, but multiple panels have issues that will cause readers, especially those with knowledge of bacterial pathogenesis, to get hung up on technicalities, accuracy, statistics, missing genes and nomenclature. Would the relocation of this figure to the end of the manuscript be better? It would reflect better reflect the abstract and there would be fewer distractors before getting to the meat of the paper. A list of issues with Figure 1 is presented below:

We do appreciate this comment and we did think about writing the paper in this way. However, the current presentation is an accurate description of how our discovery emerged, starting with initial identification of the grey colony type. Also, if we don't begin with this observation, it becomes difficult to explain why we decided to investigate transposition. Given that no other reviewers raise this point, and that order of presentation can be a subjective preference, we would like to keep the running order of the experiments as in the original manuscript.

- Fig. 1a – the grey colonies are likely mucoid colonies – mucoid and non-mucoid phenotypes are observed in Rakovitsky et al., 2021 (doi: 10.1093/ofid/ofab386), Hu et al., 2020 (doi: 10.2147/IDR.S230178), and reviewed in Shan et al., 2021 (doi:

10.1016/j.micres.2022.127057). Has a string test been done? You could do this easily (Kon et al., 2020 DOI: 10.1186/s12866-020-01971-9). I find the failure to better characterize these grey colonies underwhelming.

We would like to stress that grey colonies are not mucoid. In fact, they are exactly the opposite. The grey colonies produce very little or no capsule (Figures 1h and S1d). Conversely, the papers the reviewer mentions describe mucoidy due to capsule over production. Hence, a string test wouldn't be appropriate here. With respect to depth of characterisation, we wonder if there is a misunderstanding, since we feel we have assessed the grey colonies extensively. To summarise, we have experimentally determined that grey colonies are more adherent, produce much less capsule, lose natural competence and are less mobile.

- Fig.1b - There are discrepancies in the genomic representation of the K-locus. The authors cite Kenyon & Hall which illustrates the K-locus however, they do not include the AB5075 strain. Upon inspection, Senchenkova et al., 2015 (doi:10.1016/j.carres.2015.02.011) describe the K-locus of the AB5075 strain that the authors depict, however the loci do not match. In the present work, itrA1 is absent between gtr52 and qhbA/weeH. This work should be cited.

We cite Kenyon & Hall since this gives a comprehensive description of the K-locus and how it varies between strains.

We agree that nomenclature for K-locus can be confusing, with genes having different pseudonyms that are differently used in the literature. Importantly, the itrA1 gene is not absent in our figure. The reviewer has assumed the gene we have labelled as weeH is equivalent to qhbA. This is incorrect, weeH is equivalent to itrA1. We refer the reviewer to the Senchenkova et al paper that they mention, which states "Just upstream of qhbA, there is a homolog of itrA1 (weeH)". This is similarly stated in the Kenyon and Hall paper that we cite "ItrA1 is 93% identical to WeeH from *A. venetianus*".

Kenyon & Hall describe the qhbA gene being homologous to weeJ however, the authors label this gene as weeH. This is confusing.

We think the reviewer has misread the Kenyon and Hall paper, which states "qhbA, qhbB and gdr genes... share over 80% sequence identity with WeeI, WeeJ and WeeK... respectively". Hence, qhbA is homologous to weel, not weeJ.

We have added the pseudonyms for each gene to our figure legend. We have also explained that we use the "wee" nomenclature because this has been mapped onto the equivalent enzymology in other bacterial species (doi: 10.1016/j.abb.2013.05.011).

We have added the requested citation.

Also, how many promoters are active in this locus? Are we looking at an operon? This needs clarification.

With respect to operon structure, this hasn't been systematically determined in the literature. In our own work (unpublished) we have used cappable-seq and term-seq to

map global transcription start and termination sites in *A. baumannii*. These data are the basis of another project developing in our lab. Even so, we are happy to share observations pertinent to the reviewer's question. Our results indicate that the K-locus contains many internal promoters, as is often the case for sections of horizontally acquired DNA. There is a promoter upstream of the first gene, *gna*. This is followed by several intragenic promoters and, subsequently, a terminator downstream of *mnaB*. We suspect most *gtr52* transcription initiates from the TSS highlighted by an asterisk. Clearly, based on the complicated pattern shown below, determining the full transcriptional structure of the K-locus would require a substantial effort. In particular, it would be necessary to determine which transcripts are coding and how efficient termination downstream of *mnaB* is. For the current study, the key point is that insertion of ISAb13 prevents *gtr52* expression, which is shown clearly by our RNA-seq data in Figure 1b.

Note that only RNA ends that might be related to *gtr52* expression are labelled above.

- Fig 1c and Table S1 - Why are only some differentially expressed genes highlighted? Was a threshold level of change used? If so, what was it? I assume the title in Table S1 should be Log₂ fold change? Please fix this. Was Log₂ fold change the cut-off? If so, please state clearly.

We have corrected the table to Log₂. The cut-off was a Log₂ value of 2. With respect to P-value, the cutoff was 0.05 (although note we show Log₁₀ of the P-value). All of the significantly and differentially expressed genes are provided in the table and highlighted in the figure (although not all of the gene names are shown in the figure, only those we mention in the main body of the text for easy reference). We have also added details of cutoffs used to the figure legend. Note that data for all genes is now provided in the source data file.

Given results in panels f-h, other genes involved in motility or biofilm formation are likely to be differentially expressed egs. BfmRS, which regulates EPS and pili

production, OmpR/EnvZ bacterial envelope stress – but these are not seen in the figure or the Table – why not? Are these genes encoded by the large plasmid (83 kb) that is not stably maintained (see whole genome sequencing section L350)?

The reviewer is assuming that genes like *bfmRS* and OmpR/EnvZ will be differentially expressed. The data show this is not the case and this is why they are not shown. We also suggest that the logic the reviewer uses here is, perhaps, incorrect. The changes in biofilm production, motility, etc arise because the capsule is lost (e.g. it's known cells lacking capsule are more adherent). Whilst these phenotypes can also be controlled by the regulators mentioned, this in no way means that observation of such phenotypes must be linked to differential expression of the regulators. In other words, just because regulator X controls phenotype Y, that doesn't mean changes in phenotype Y result in differential expression of regulator X. Indeed, most regulators are not controlled at the expression level, but on the basis of a signal (e.g. cAMP for CRP) being present or absent.

In this regard, complementing the mutant with the ISAb13 with an exogenous copy of the locus would help determine which phenotypes are caused by the IS element and which are artifacts of large plasmid loss. This part of this study is not methodical.

We think there is some confusion here. The reviewer seems to be implying that the large plasmid being lost could give rise to phenotypes associated with ISAb13 disruption of the K-locus. For this to be the true, the plasmid would have to be lost specifically from cells carrying ISAb13 in the K-locus, and not WT cells. This is not the case, the plasmid is equally prone to loss from wild type and grey cells; there is no link between grey colonies, their phenotypes, ISAb13 and the plasmid. Also note that any plasmid loss would impact only a small number of cells in the population. The assays we have used measure batch phenotypes of the whole population. Whilst considering the reviewer's point, we also realised that the lower number of reads mapping to the plasmid, which we assume is due to plasmid loss, could have another explanation. Specifically, if the large plasmid is isolated less efficiently, when DNA is prepared for sequencing, this would result in a similar effect. We have added this detail in the methods section.

Last, we would like to draw the reviewer's attention to the parallel study of Whiteway *et al*, who makes a clean knockout of the K-locus and show very similar phenotypes (<https://www.biorxiv.org/content/10.1101/2024.02.15.580542v1>). This provides strong independent validation of our observations.

- Fig. 1e: The growth defect exhibited by the grey mutant appears restored after 24 hours. Has this strain reverted to the wild-type phenotype or obtained a second site suppressor? What is happening here? If the IS element has jumped again this would be relevant to include.

We did test this at the time and the IS element has not jumped again (we, like the reviewer, thought it may have done). The associated data are shown below. Briefly, we used oligonucleotides for PCR that amplify DNA including the site of ISAb13 insertion. Wild type cells give the "red" product and the cells with ISAb13 at the region of interest give the "blue" product. Cells were taken for analysis by PCR at the start of the

experiment (T_0) and after growth overnight (O/N). We also plated the cells to check for the grey phenotype and this had similarly not been lost.

Minor Comments

Abstract L38-40 “For example...” is an odd way to start a sentence

I think we’ll have to agree to disagree here. “For example,” is a common way to start a sentence.

L48 There is a comma missing before traits. Add hyphen “horizontally-acquired”

The comma has been added but we have left “horizontally acquired” unhyphenated to be consistent with other instances. We appreciate that the hyphen may be grammatically correct but, in general, people don’t tend to hyphenate this term.

L57 Add Picker et al., 2016 doi: 10.3390/genes7120112 to support that H-NS is conserved throughout the gamma Proteobacteria

The paper mentioned is a review of H-NS family members and their regulation of virulence genes in Shigella species. The review doesn’t seem to discuss the conservation of H-NS in gamma proteobacteria. As such, we would prefer to keep the original citation that presents a comprehensive phylogenetic analysis of H-NS.

L88 What is the position of the K-locus in AB5075 strain? The position of the two other ISAbA13 is provided. Add the position at the K-locus too.

The K-locus is between positions 3907256 and 3931673 of the chromosome. This detail has been added in the legend to Figure 1.

L94 Change tex. 45-fold effect is log2 of -5.94

We have altered the text.

L133 What is the nature of the *hns* mutant? Table S4 says T26 transposon in *hns*, but details are missing. Where is the transposon? Does this generate a true null, or is some part of the HNS protein made? This is important as N terminal truncations of HNS display dominant negative effects on other HNS paralogs and *Acinetobacter* has a *stpA* gene. Thus, there is scope for an HNS truncation to interfere with StpA binding. Please address the nature of the *hns* mutant that you've used.

The transposon inserts at base pair 211 of the *hns* gene, causing a frame shift. This detail is now added in Table S4 and briefly mentioned in the methods section. The site of insertion (red triangle), in context of the H-NS protein sequence, is shown below. We have also done native Tn-seq with a full H-NS deletion and the results are identical to those obtained using *hns::T26* (see our response to an earlier query above and the associated native Tn-seq results). These data, to be part of a future publication, rule out the possibility raised by the reviewer.

As noted above, *A. baumannii* does not have StpA.

MKPDISELVSVEELKRLQEEAEALIASKKDQAIEDAYNQIIEIAENVGFSVEQLLEFGAQRKKITRKSVEPRYRNKNNAEETWIGRGKQPRWLV

AEIEKGAKLEDFLI-

A comment about this section generally – What effect does temperature or osmolarity have on transposon capture? If H-NS is more tightly bound at 30°C wouldn't you expect more capture at these ambient temps than at host temps? It would be odd for a pathogen to exhibit heightened mutability outside of its host – normally we consider pathogens to become more genetically variable in the host environment.

We have looked at this in a very limited way, and the data, if published, will be part of a future study. There do seem to be differences induced by osmolarity (we haven't looked at temperature) but this is a very preliminary result.

Is StpA, the well-established H-NS paralog, involved in IS element capture? If the *hns* mutant is a true null then the answer is no, based on your data, but the lack of details makes it challenging to discern, emphasizing the need to include the nature of the *hns* mutant used.

We don't know at this stage if StpA is involved. We are currently doing experiments to test this in *E. coli*. As noted above, *A. baumannii* does not have StpA.

L145 “Two plasmids... contain regions of elevated transposition”- which two? Be specific. If it is the two largest ones is this due to the amount of DNA being higher i.e. larger plasmid, more DNA, more likely to be hit. Please clarify this point.

In our defence, we do direct readers to Table S3 at the end of the sentence, where the details are presented. In full, the sentence reads “Two of the plasmids, and 7 of the prophage, contain regions of elevated transposition dependent on H-NS binding (Table S3).” We have not listed the plasmids since we would also then need to list the 7 prophage, which is cumbersome. Importantly, we are talking about transposition dependent on H-NS binding, not total transposition. Hence, plasmid size is not really the key issue here, it’s the presence of H-NS that’s important.

L165 It is unclear how IS elements that hit the essential genes are being counted? Wouldn’t these be eliminated as these mutations would be lethal? Are you picking up signatures from dead bacteria? Or are the insertions in positions within the essential gene that hobble the gene without knocking it out completely? Some further explanation is warranted.

We believe we are picking up several things i) signals from dead cells ii) signals from cells that are still alive but unable to replicate iii) insertions of ISAb13 that allow the gene to retain some functionality. We have added a little more text for explanation.

L197 What is the purpose of the statement about the weaker secondary diagonal? This is never mentioned again – it is almost an orphaned thought.

The text on line 197 states “a weaker secondary diagonal indicates interactions between opposing chromosomal arms”. We come back to this on line 247 and state “our 3C-seq analyses show that each arm of the *A. baumannii* chromosome interacts with the other (Figure 4d)”. We then present an analysis of correlation between transposition frequencies, at equivalent sites on each chromosomal arm. Hence, there is a purpose to the statement, and we do come back to it on the following page.

Is it weaker? That is not abundantly clear. We think there must be some confusion here, since the secondary diagonal is clearly weaker (see below). Perhaps the reviewer was confused by what “secondary diagonal” is intended to mean? We have added an extra line of text to try to avoid confusion. Also see the below schematic.

Furthermore, doesn’t the sentence in L199 contradict this statement? This causes confusion.

There is no contradiction, if we understand the reviewer correctly. The following sentences concern the impact of H-NS-39 on contact frequencies at 10 kb resolution. This is unrelated to our observation that the two arms of the chromosome interact.

L205 It would be good to tie this observation back to Vicky Lioy's work in *E. coli*, where H-NS makes short-range interactions (Lioy et al., 2018; <https://doi.org/10.1016/j.cell.2017.12.027>)

We do already make this link just prior to the position the reviewer suggests (see the sentence "Previous chromosome conformation capture studies in *E. coli* also reported minimal effects of H NS on 10 kb resolution contact maps⁵¹"). Hence, we have not altered the text. We would like to make the point here that, whilst broadly consistent, our observations, compared to those of Lioy *et al* for *E. coli*, are subtly different. In Lioy et al, it is shown that short range interactions increase when H-NS is deleted. In our paper we have not tested what happened when H-NS is deleted, instead, we have inhibited DNA bridging by H-NS. When bridging is inhibited, short range interactions are reduced. Our interpretation is that removing H-NS completely allows new interactions, not involving H-NS, to occur. Conversely, inhibiting bridging by H-NS will prevent local interactions between H-NS molecules but, because H-NS is still bound to the DNA, new interactions are probably blocked.

L210 In this section and throughout the paper it is unclear how the HNS39 allele was expressed in vivo – this is essential information and must be included.

We have added text to the first section of the methods.

L211 Change the text to "Our observations are consistent with HNS 39 impeding the ability of native H-NS to bridge, under the conditions used or under the expression levels used in vivo. This would be more accurate and relates to concerns about ratios of HNS and HNS 39 raised in the major comment #1.

We have added "in the conditions used here".

Section starting L221 – This section would benefit from a more concrete concluding statement that explains which of the two scenarios pertaining to pre- post excision, raised on L 223-225, is more likely based on your data.

We have altered the text.

L260 Is IS903 an IS5 derivative? What about InsH3 (L271)? This is worthy of comment, given paragraph 1 of the Discussion.

Yes, and this detail has been added.

Discussion points to consider for inclusion –

i)H-NS increases *Acinetobacter baumannii* competence – Le et al., 2021 this is relevant in paragraph #2 where you discuss horizontal gene transfer and H-NS.

We have added this discussion point, but at the end of the first introductory paragraph, where we felt this a better fit.

ii)Other NAPS – are they involved in this type of process? Is there any evidence?

Most likely they are, but the evidence at the moment is very limited to a small number of NAPs and a small number of transposition events.

iii) What is the potential for involvement of StpA in this phenomenon?

We imagine that there is some impact in *E. coli*, and we plan to test this. Note that *A. baumannii* does not encode StpA. We do note the potential role for factors that modulate H-NS activity already, so haven't added any further text.

iV) Environmental parameters that affect H-NS DNA engagement, are likely to affect IS element capture too, right? How does this relate to pathogen genetic plasticity – there's scope to discuss this.

We agree, and have started some experiments to investigate this in more recent work. In this case we haven't added any extra text.

Methods

L366 while Whiteway is cited, the original methods are described in Kon et al 2020 (DOI: 10.1186/s12866-020-01971-9) Replace or add this citation.

The requested citation has been added.

L376 The procedure was first described by Tomarus et al 2008 and Allen 2020. Add or replace these citations.

We do not think that these are the first descriptions of the procedure. The Tomarus paper refers to a 2003 study, by the same first author, that itself refers back to a Roberto Kolter methods paper from 1999. Of course, all of the methods are based on binding of crystal violet to bacterial cells, which is a very old observation. In the case of our paper, we did adapt the procedure from our own prior study and the study of a colleague. Hence, we cite these two papers.

L384 Please include how the cells were grown immediately prior to RNA-seq library preparation. These details are important.

Growth conditions are provided in the first section of the methods.

References

L691 – Van der Valk citation #50 is incomplete. This is a repeating problem in this section.

We have checked the references throughout to correct errors (note that these changes do not show up as highlighted text since the changes are instigated by the reference management software).

Figures

Figure 1 – see major comment #3

See response above.

Figure 3b What are the genetic loci shown? Add locations or some descriptor.

Locations have been added to the figure legend.

Figure S2a What are the genetic loci shown? Add locations or some descriptor.

Details have been added to the figure legend.

Reviewer #2 (Remarks to the Author):

Reviewer #3 (Remarks to the Author):

The manuscript by Cooper et al. reveals a new role of H-NS in regulating transposition. Unexpectedly, the authors show that, rather than insulating the DNA from transposable elements, H-NS targets transposition to DNA regions that it is known to silence, such as virulence genes and prophages. This study uses mainly NGS-based technologies to investigate H-NS effects on transposition of ISAb13 element in *A. baumannii*, which can cause serious infections. The results demonstrate that H-NS can promote the insertion of transposons by physically capturing them and that the capture is mediated by DNA bridging activity. The authors also provide explanations of apparent discrepancies between their data and those published previously (mostly with the *E. coli* H-NS) and data analysis consistent with a notion that *E. coli* H-NS favors transposition into genomic loci it silences.

The presented experimental data are convincing, the manuscript is well written, and the logic is clear. Overall, these findings are significant for our understanding of bacterial evolution and the role of NAPs therein and would be of interest to broad audience. However, this reviewer has a number of suggestions that may help to make the manuscript more accessible.

Major:

1) A major conclusion is that H-NS manipulates transposition to maximise favorable evolutionary outcomes for the host, by preferentially targeting transposons (at least this one) into foreign DNA that is silenced by H-NS. I think that a separate emphasis on plasmids and prophages, which are ultimate foreign DNA, is not warranted.

We understand the reviewers point here but have to disagree. We think making a distinction between plasmids/phage (which have the potential to move to a new cell) and other sections of horizontally acquired DNA (which do not) is important because of

the implications for transposon dissemination and plasmid/phage inactivation. We would like to emphasize that this distinction only takes up 9 lines of the results section and a short paragraph in the discussion. This doesn't seem unreasonable.

While, as stated in line 302 "by facilitating transposition into prophage or plasmids, H-NS provides a route via which transposable elements can spread horizontally", how many H-NS binding sites are located in prophage/plasmids? If the binding sites in prophage/plasmid are not the majority, the authors cannot single out prophage or plasmids as "special" transposition targets.

We don't understand the logic here. In our view, it isn't important if H-NS binding in phages/plasmids represents the majority of H-NS binding genome-wide (i.e. compared to the chromosome). More important is that i) the majority of phages/plasmid in our experiment are bound by H-NS and ii) that this has important implications worthy of discussion.

Overall, we appreciate the reviewers point. However, given the important implications for phage and plasmids, and that only a very small part of the paper is dedicated to this topic, we don't think our presentation is unreasonable.

This information should be discussed using the ChIP-seq data in the manuscript.

We do already do this, in our view, with the sentence "To understand the effects of H-NS, we examined our ChIP-seq and native Tn-seq data. Two of the plasmids, and 7 of the prophage, contain regions of elevated transposition dependent on H-NS binding".

2) Any explanation for the high number of insertions in the H-NS unbound region? Figure S2a shows a high peak of insertion sites in essential genes of wt cells. What is the essential gene?

We would like to make two points in response:

1) The gene in question is homologous to *E. coli* *bamA* (encoding a subunit of the BAM complex, also essential in *E. coli*) and the insertion is very near to the 5' end of the gene. This may not completely inactivate the gene.

2) It's important to point out that a "high peak" doesn't mean a high number of independent insertions. That particular insertion may have occurred only once, and then been inherited by subsequent generations as cells divide. The details are in the methods section, but this is why we don't take read depth into account (i.e. each peak counts only as one insertion). More important is the density of insertions (for which we use the "insertions per kb" metric in Figures 2 and 3). The reviewer can probably see, by looking at Figure S2a, that the density of insertions is much higher in the regions bound by H-NS.

3) H-NS-39, a truncated variant of H-NS, has been used to argue that the DNA bridging activity of H-NS is responsible for transposon capture. However, the *in vivo* influences of H-NS-39 on DNA bridging are limited. How many sites are influenced?

We probably haven't explained this clearly enough. Short-range interactions are extensively altered throughout the chromosome by H-NS-39. This is clear in the matrix shown in Figure S4b; the thin region of red coloration along the main diagonal indicates loss of short-range interactions. It's difficult to quantify how many sites are influenced because the reorganisation is so widespread. For instance, if the reviewer compares the two panels in Figure S4c, they will see that the lost loop is accompanied by extensive subtle changes throughout the region. It's impossible to divide this up into specific "sites" because there is no repetitive feature of the contact map that changes; we just see extensive local rearrangements. We have added extra text to the relevant section for clarification.

Figure 4 and S4 show only 5 sites and all these sites have low H-NS occupancy.

The examples we show in Figures 4e and S4c-f are chosen because they are nice instances of obvious loops being lost. To determine how the 3C-seq signal for each loop aligns with the H-NS ChIP-seq, it is necessary to look at the data as outlined by the dashed lines below (for Figure S4c). This instance indicates H-NS binding at the base of the loop. In other cases we see H-NS at the centre of the loop or, at the base, but only on one side. We don't exclude the possibility that some changes, particularly those which are more subtle, could be indirect consequences of lost H-NS bridging (e.g. if an H-NS bound region is reorganised it seems likely that there will be effects on adjacent DNA).

An assumption here is H-NS-39 abolishes DNA bridging.

We agree that it's difficult to prove bridging unequivocally in vivo (beyond the 3C-seq evidence). That said, the assumption is based on sound biochemical data in Figures 4 a and 4b, as well as prior data from the van der Valk paper. We think this is a reasonable assumption.

What if H-NS binds to the transposase to increase transposition and 39 peptide blocks this interaction?

We considered this during our project and tested for a direct interaction between H-NS and the transposase using a 2-hybrid assay. The result, as an excerpt from the first author's PhD thesis, is shown below; we find no evidence of an H-NS-transposase interaction. That said, we do not unequivocally rule out the possibility. For instance, H-NS and the transposase could interact at a very specific stage of the transposition process that cannot be captured in a 2-hybrid assay.

Also note that we do detect an H-NS-H-NS interaction when using this assay, but only on the basis of plate phenotypes, not in LacZ assays (we don't understand this, but have seen this before with other proteins).

4) The argument for “Transposition events are likely captured after ISAb013 excision” needs to be elaborated. For example, line 237 states “This contrasts sharply with transposition frequency”, but the meaning of this observation is not very clear.

To put our text into full context, we state “The bar chart in Figure S5b shows the number and location of contacts made by each ISAb013 copy. In both cases, most interactions occur within a ~20 kb region surrounding the element; contacts with chromosomal regions elsewhere are both infrequent and uniform (Figure S5b). This contrasts sharply with transposition frequency (Figure S5c).”

In other words, the existing copies of ISAb013 only interact with DNA that is very close by, not distant regions that are transposition hotspots. If ISAb013 was being pre-captured by transposition hotspots, perhaps due to long-distance interactions mediated by H-NS, we would expect to see this interaction in 3C-seq. What we actually see is that the existing copies of ISAb013 are amongst the most-poorly interactive parts of the genome in 3C-seq; they keep themselves to themselves. Hence, ISAb013 is more likely captured after excision.

How to make sense of “proximity to an existing copy of ISAb013 could impact transposition independently of H-NS” since the two indicated regions (especially ISAb013 copy 1 area) do not have significantly higher peaks than others in Figure S5c. With that being said, consider moving Figure S5 to the main figures.

Let us try and paraphrase. Our data suggest that H-NS is not exerting its effect on transposition frequency by bringing transposition hotspots into proximity with existing IS*Aba13* copies. However, that doesn't mean that being close to IS*Aba13* doesn't result in higher levels of transposition (Figure S5 is irrelevant to this argument).

By means of analogy, this is like making the observation that people who drive bright red cars are more likely to be involved in accidents (with the car colour being equivalent to H-NS and the accident being a transposition event). A hypothesis might be that this is because people with bright red cars drive on busy roads more often (busy roads being the equivalent to an existing IS*Aba13*). To test this, you measure how often red cars drive on busy roads compared to other cars. You find having a red car (i.e. being bound by H-NS) makes no difference. The conclusion is that having a bright red car (H-NS) does not make you more likely to drive on busy roads (i.e. be in close proximity to existing IS*Aba13*). However, at the same time, this does not mean that driving on busy roads (i.e. being near to IS*Aba13*) has no influence on accident frequency (transposition) independently of car colour.

Minor:

1) Are the positions 1.764 Mb and 3.863 Mb covered by H-NS? How does the IS*Aba13* dissociate from H-NS after capturing? Any thoughts on this?

See below, both are in H-NS bound regions. If we look at the instances of IS*Aba13* in the chromosome, we can see that H-NS is bound to the extremities of the element. Perhaps H-NS bridges between the chromosome and IS element ends and doesn't need to dissociate.

2) Does *A. baumannii* have another homolog of H-NS? *E. coli* has H-NS and StpA.

We have covered this extensively in our response to reviewer 1. To summarise, *A. baumannii* does not have StpA. The organism does encode an H-NS like protein (nearly identical to chromosomal H-NS) on one of its three plasmids. The plasmid H-NS has no impact on global patterns of transposition (see data showing this in response to reviewer 1). We believe that the plasmid H-NS could be having some more subtle, and very interesting, effects. This is currently the topic of a separate project in the lab that we hope will be the basis of a future publication.

3) Line 70: reference 30 is not the latest list. Please use the WHO bacterial priority pathogens list, 2024 (<https://www.who.int/publications/i/item/9789240093461>).

This list was updated after submission. We have added the suggest reference (and have also kept the original one, since the new addition is not a formal publication).

4) Fig. 1a and Fig. S1a are basically the same. If anything, Fig. S1a can show more information.

We have presented the figures in this way to try and avoid confusion. We feel that introducing a third colony phenotype in the main figures isn't necessary for the casual reader (only those more familiar with *A. baumannii* will be aware of the additional colony variant, that is unrelated to this work).

5) Line 93: "ISAb13 appears upregulated". What is the read count for ISAb13? There is no information in the Method describing the read-count cut-off for differential gene expression analysis.

The \log_2 fold change is in Table S1 and we now include information about cutoffs in the figure legend. Data for each individual gene are now available in the source data file.

6) Line 97: "mediated by a type IV pilus requiring pilA". Citation?

We have added two relevant citations.

7) Line 123: Fig. 2b cannot clearly show that "transposition is biased towards non-coding sequences". Another pie chart that shows what proportion of the genome each class occupies needs to be shown, or % genome can be added inside the slices..

We understand the point. Rather than add another pie chart or %s, which we think could be confusing, we have added some text to the figure legend.

8) Fig. 2c: How is the correlation coefficient of Fig. 2c calculated? What is the gene region of the expansion?

The value shown is the Pearson correlation coefficient. This detail has been added to the various figure legends. We didn't understand the second part of the question as exact co-ordinates of the expansion are provided in the figure (approximately 2.525 Mbp to 2.675 Mbp). Obviously, this is too large to list individual genes if that is the query.

9) Line 150: should it be "targeting transposable elements in prophage and plasmids"?

No, because we mean that H-NS binding on plasmids is targeting transposable elements to these locations. This is subtly different to the reviewer's suggestion.

10) Line 208: define hairpin loop

To avoid confusion, we have removed hairpin ("loop" seems self-explanatory).

11) Line 328: “Lysogeny Broth”.

We have made the alteration.

12) Line 331: Italicize “A. baumannii”.

This is intended, since the sub-heading is italicised A. baumannii is not.

13) Lines 355-356: please cite publications for these bioinformatic tools. Also, how are the reads trimmed? The reviewer recommends the authors check out “minimap2” for nanopore reads alignment in the future as minimap2 has replaced bwa-mem for long-reads alignment (<https://github.com/lh3/bwa>).

We have added the citations. Noted re minimap2.

14) Lines 361-364: please consider using the name p1AB5075 for this 83.61 kb plasmid. If the p1AB5075 was lost, why can it be assembled from Illumina sequencing data? If only part of the population lost p1AB5075, the authors should make efforts to get more sequencing depth with nanopore and assemble this plasmid. The assembled plasmids should be included in the NCBI Genome Assembly submissions (GCA_036602705.1 and GCA_036601155.1) to make a true “Complete Genome” assembly level. The p1AB5075 has antibiotic-resistance genes, which can be used for testing its presence.

We have added p1AB5075 as requested. Note we do not say we can assemble the plasmid from Illumina data. We say we can see Illumina reads mapping across the entirety of the full 83.61 kb p1AB5075 described in previous work. Hence, we know that the complete plasmid is present. The number of illumina reads mapping to this plasmid is lower than the number of reads mapping to the chromosome and other two plasmids. Hence, we assume it is being lost from some cells. Thanks for the tip re antibiotic resistance genes, we’ll use this in the future.

15) Line 369: an extra period mark.

This has been corrected.

16) Line 386: “as described previously”, insert citation?

This has been done

17) How many repeats for RNA-seq and ChIP-seq?

Two, this detail has been added.

18) Lines 398 and 483: 50 bp windows and 99 was selected as the threshold. If the authors had a good reason for the cutoffs, please explain it in the Method. The readers will benefit from it.

We determined reasonable cutoffs following visualisation of the data and comparison of transcribed vs non-transcribed (or H-NS bound vs unbound) regions. This detail has been added.

19) Lines 406 and 408: “dH₂O” is not a common word for everyone to understand. Please use the full name.

We now state “deionised” H₂O.

20) Line 410: “5 ul 0.5 M EDTA” doesn’t make sense because no total reaction volume is provided.

This has been altered to include the total reaction volume

21) Are the coverages for detected sites of transposition reasonable? Are they all higher than 30?

We count all sites of insertion, regardless of read depth. This is necessary to detect the rarest insertions (e.g. in essential genes) that are unlikely to get amplified in the population by subsequent cell division. We do, however, throw out reads where the genomic sequence flanking IS*Aba13* is not at least 20 nt in length. This minimises the possibility incorrectly mapping genuine IS*Aba13*::chromosomal DNA junctions. Also note the inclusion of a hemi-nested PCR step, that ensures we only capture genuine IS*Aba13*::chromosome junctions (illustrated in the new Figure S7).

22) Line 459: ”read depth 1 position”?

I.e. one base pair downstream.

23) Line 544: “dividing the value in each bin my the”?

Corrected to “by the”

24) Lines 552, 553: editing “heparin HP1”, “against”).

This has been corrected

25) Line 569: what is the reaction buffer?

See the text “Beads were washed twice with, and then resuspended in, 16 µl of 10 mM Tris-HCl pH 7.9, 0.02 % (v/v) Tween-20, 1 mg/ml Acetylated BSA, 1.25 mM Spermidine, 12.5 mM MgCl₂, 6.25 % (v/v) glycerol, 1.25 mM DTT.”. This is the reaction buffer.

26) Line 792: and grey variant cells.

This has been corrected

27) Line 809: depth

This has been corrected

28) Which cells are used for native Tn-seq? Wild-type or the grey variant. If wild-type, please show the depth of 1.764 Mb and 3.863 Mb locations for comparison.

We used wild type cells. As discussed above, we have not used read depth because this is not representative of transposition frequency. This is particularly true for the two starting locations; because transposition is a comparatively rare event, the vast majority of cells still have ISAb13 in these locations. The comparison the reviewer requests is shown below, with the y-axis zoomed out to show the full read depth for one of the native ISAb13 copies (top) and the zoomed in (bottom) to better see new transposition events. Note the change in y-axis maximum in the top right of each image.

29) Figure S2b, how were insertions per kb calculated? Is the method different from Fig. 3d?

It's the same method as in 3d. Each site of insertion counts once, regardless of read depth, then we determine how many insertions we see per 1 kb of the genome.

30) Figure 4a, the surface for DNA binding is also dark green. Is the dimerization surface pale green?

The two regions of the protein can be differentiated because the DNA binding region is shown as a circle whilst both the dimerisation and multimerisation surfaces are shown as semi-circles. This is noted in the figure legend. In responding to this comment, we also realised we had drawn our schematic incorrectly (showing H-NS-39 blocking the dimerisation rather than the multimerization interaction). This has also been fixed.

31) Figure 4d can be moved to supplement. Fig. 4e, it's not easy to see the difference. Please indicate the actual values for the discussed part. The same goes for Figure S4.

We would prefer to keep 4d in the main body of the Figures since it shows the first contacts maps published anywhere for *A. baumannii*, so is important in this regard. The Figure also illustrates the lack of an obvious effects due to H-NS at the 10 kb scale.

32) Figure S4b legend: "b. a. Changes"?

This has been corrected.

33) Why are the correlation coefficients in Figure 5a and Figure 3a different? Maybe use "correlation coefficients (r)" in the legend.

These are different (albeit very similar) experiments. Hence, the correlation coefficient is also slightly different. Whilst both strains are wild type, the cells in the latter experiment also carry the empty pVLR1Z plasmid. We have also altered the legends.

34) Figure 5c, is this wt or H-NS deletion cell?

Wild type (note the label to the left of each heatmap)

35) Figure S6a, what is the correlation coefficient?

This has been added (it's 0.44).

Reviewer #4 (Remarks to the Author):

The paper by Cooper and colleagues focuses on a largely neglected function of the nucleoid-associated protein H-NS in bacteria: its ability to influence target site selection by transposable elements. H-NS, a major nucleoid structuring protein in gamma proteobacteria, binds to hundreds of sites in the bacterial chromosome, silencing the expression of genes relevant for host invasion or environmental adaptation. This paper shows that sites bound by H-NS constitute hotspots for the insertion of ISAb13, an IS5-type transposon in *Acinetobacter baumannii*. Interestingly, the targeting of H-NS-bound sites by the transposon is mediated by H-NS's DNA-bridging activity; a finding that lays the ground for future studies of the mechanism involved. The paper also discusses preliminary evidence that suggests a similar trend in *E. coli*, pointing to the generality of the phenomenon.

While the participation of H-NS in transposition was recognized prior to this work, its role in target capture has never been documented with the genome-wide depth achieved in this study. Given the relevance of H-NS-controlled processes in *A. baumannii* pathogenicity, the phenotypic diversity resulting from the insertional disruption of H-NS-regulated genes is an important aspect of the biology of this medically significant bacterium. The data in the paper are clearly described and convincing overall. The clinical implications of the work can be expected to attract the attention of a wide readership. I only have a couple of comments that the authors may want to consider in revising the manuscript.

Thanks for the positive comments.

1. An obvious shortcoming of the "Native Tn-seq" procedure used in this study is the potential bias in the profile of transposon insertion sites that might result from differential growth rates of the insertion mutants.

We agree, and there is also the issue that any insertion arising early during growth of the population will be over-represented compared to those that arise late on (assuming no bias in growth rate is introduced by the insertion). To get around this issue, the best we can, we don't take read depth for insertions into account. For example, an insertion that giving a growth benefit, or arising early during growth of the culture, will be amplified by cell division. Conversely, an insertion in an essential gene will never be amplified by cell division (unless the gene is not completely inactivated). In our analysis, even if the former insertion generates a much greater read depth, it still counts as only one insertion, just as an insertion in an essential gene would.

The method description (line 400) indicates that genomic DNA was extracted from a mid-exponential phase culture, sub-cultured from an overnight culture (lines 402-403) presumably inoculated from a single colony. It is highly likely that most insertion mutants identified in the subsequent analysis were already present in the starting colony (which typically contains approx. 10^9 cells). Their undergoing several divisions prior to cell harvesting may significantly affect their relative representation. The authors should discuss this point and provide more details on how these steps were performed: were the biological replicates inoculated from separate single colonies? What fold-dilution was used to sub-culture the overnight culture?

We didn't use single colonies, we inoculated overnights from glycerol stock scrapes prepared from liquid cultures, so far fewer cells than in a single colony. After overnight growth, we sub-cultured again with a 50-fold dilution of the overnight.

These details have been added to the methods section.

In the methods section describing strain handling (line 326), the authors state: "To avoid the use of sub-cultures, inoculated liquid media was left at room temperature overnight before being transferred to 37°C, with shaking, until cells reached the required growth phase" (lines 328-330). For which experiment was this method used?

Apologies, this does require further clarification and we have altered the text. This was done on all experiments except native Tn-seq (where we wanted to maximise population diversity). The reason is that we try to avoid passaging hns mutants too many times during an experiment (e.g. by plating out from a glycerol stock, setting up an overnights, and then subculturing). This is because, in *E. coli* at least, cells can acquire mutations that suppress the hns effect.

Although eliminating the sub-culturing step is a good idea, adding a room-temperature static growth step may introduce additional bias; for instance, mutants adhering to the tube walls may be left behind. Why not perform the entire culture expansion from colony to the "required growth phase" in one step, with shaking, at 37°C? This would minimise the number of cell divisions without introducing unnecessary changes in growth conditions.

We're a little confused here as the comment above, also regarding the use of single colonies, seems to imply that this is not ideal. If the reviewer is worried about detection

of transposition events, that occurred in the colony, it seems logical that sub-culturing might be helpful.

We're also not sure we see a problem. Clearly, when we look at ISAbA13 positions, these will have all arisen in the past, some many cell cycles ago, and others more recently. We're interested in the overall pattern, and how this relates to the presence or absence of hns, so does it matter when the transposition event occurred?

Note, however, that the most rigorous way to perform this analysis, which the authors might consider for future work, would be to engineer a strain carrying an inducible transposon and prepare the DNA library shortly after induction.

We do now have such a strain, although we do think it is nice to look at naturally occurring transposition.

2. The authors may want to elaborate on their vision of how DNA-bridging by H-NS could enhance target capture by the transposon. Plausible scenarios can be inferred from the notion that a crucial step in target capture involves transposase-mediated bending of the DNA. It is conceivable that factors promoting bending at specific sites might facilitate transposon insertion at these locations. Could the apexes of DNA loops or plectonemes stabilized by H-NS-mediated bridging provide such favorable bends?

This is possible. We also wonder if H-NS might interact with the ends of the loop as in the schematic below. One of the reasons for suggesting this is that the extremities of the transposable element are more AT-rich and, at the sites ISAbA13 sits in the chromosome, we see H-NS binding signals at each end of the IS element in ChIP-seq. Obviously, this is all speculation at present.

Minor points.

3. Line 85. Add "a" to Fig. S1.

This has been done.

4. Line 116. Not clear what bias the authors are referring to.

Our logic is that, in chromosomal DNA purified from cells, most ISAbA13 copies will be in the two starting locations. PCR is unlikely to amplify every single potential template

DNA molecule during library prep. If this is the case, and only some templates get replicated in the first round of PCR, these are disproportionately likely to be those having ISAb13 in the starting positions. The bias will then be exaggerated in every subsequent PCR cycle. We have added a little extra text for clarification.

5. Line 121. Whiteway et al; insert reference number.

Now added at the end of the sentence.

6. Line 188. Insert “the” between “understand” and “effects”.

Done.

7. Line 312. Strictly speaking the term “scarless” applies to the reversibility of the ISAb13 insertion, not to the insertion itself.

We have modified the text.

8. Lines 400-447. Native Tn-seq section. Providing a schematic diagram of the step involved in the library preparation as a supplementary figure would help understanding the workflow.

This has been added as Figure S7.

9. Line 468.”...chromosome was altered to encode H-NS with a C-terminal 3xFLAG fusion”. This is a rather cryptic and unorthodox way to describe a strain construction. Please provide details.

We have added the required details and have moved the text to the “Strains, plasmids and oligonucleotides” section of the methods.

10. Line 950. Remove “a” after “b”.

This has been done.

11. Figure 1e. The filled circles plot is invisible. Consider splitting the diagram (placing active and inactivated serum plots in separate side-by-side panels).

We haven’t made this change since the raw values for the plot are now provided as source data.

12. Figures 3b, 5b, S2a, S3a right, S3b. Please add chromosome position of genes (Mb); add gene names in S2b and S3a right.

We have added chromosome positions to the figure legends. We haven’t added gene names since these are most in the format ABUW_1234 etc that are not so informative.

13. Figure 4d. What are those “empty” bins in the terminus region?

That's a good question, and we don't know the answer. We initially wondered if this was some sort of computational artefact, or resulted from a repetitive sequence to which contacts can't be assigned. Neither is true and, on close inspection, a small number of contacts do map to this region. It's interesting that, in hns- cells, this region acquires more ISAb13 insertions than other parts of the chromosome (the two starting locations of ISAb13 aside), you can see this in the lower heatmap of Figure 3a. To us, this hints at some chromosome structuring phenomenon, but that's just a guess.

Reviewer #5 (Remarks to the Author):

Reviewer #1 (Remarks to the Author):

The authors have done an excellent job addressing comments raised during our review of this submission to Nature Communications. However, four areas warrant further consideration and comment. These may be considered somewhat tangential to the main topic of the paper, but addressing these will strengthen the whole paper.

1) Does *A. baumannii* have StpA or other H-NS like proteins?

The authors indicate that this strain of *A. baumannii* does not code for StpA, but has a plasmid borne H-NS like protein, which they've coined H-NS-2 (but this protein is never mentioned in the paper, which is OK given they are working on it). Nevertheless, and importantly, other strains of *A. baumannii* do have StpA (see strain 4300stdy6542372 (Genbank UFD000000000.1). Firstly, is it possible that Genbank SSI78197.1 is the H-NS2 that the authors mention in their rebuttal? Comparing these two proteins seems more appropriate than comparing them to StpA from *E. coli* MG1655. If they are similar/the same protein (which seems unlikely based on the provided alignment), then this is worthy of comment and re-annotation. If they are different, it will highlight the plasticity of the different *A. baumannii* strains and reveal the potential for variability in the presence of H-NS and H-NS-related proteins, which are/may be required for the system being studied.

Secondly, there is no description in the discussion of StpA and how it relates to H-NS. The added sentence "this may explain why phage and plasmids encode H-NS modulating factors" only relates to their finding that H-NS2 is encoded by this strain of *A. baumannii*. But this hasn't been shared with the readers – it only make sense if you know the strain has h-ns2. The discussion would merit from some enhanced discussion of the potential/scope that StpA and other H-NS-like protein may play in the process that has been studied - transposon capture.

2) Grey colonies and mucoidy.

From our vantage, and likely others will agree, the grey colony's morphology looks mucoid. Additionally, other phenotypes displayed by this colony and characterized here, i.e., more adherence and less motility, are classically exhibited by mucoid cells.

In the rebuttal, the authors appear to indicate that mucoidy is only associated with strains that are producing capsule, but this is not accurate. Colonies can become mucoid even when the classic capsule genes are deleted. Instead, the mucoid phenotype manifests when cells are producing large amounts of EPS that gels on the cell's surface (see Lembre book chapter doi 10.5772/51213, "Polysaccharide molecules can interact with themselves or with heterologous ions and molecules to yield gels, often with multivalent cations playing a significant role in the process." Taken from above citation; Introduction, paragraph 2).

All it takes is for other polysaccharide-generating pathways to be upregulated. Are there other polysaccharide biosynthetic genes in the genome of AB5075? Are other sugars being produced in large quantities? For instance, in *P. aeruginosa*, mucoidy often results from an overproduction of alginate. This is why a string test was suggested in our initial review. It's a very simple assay to assess mucoidy and would address this point directly, eliminating guesses by readers about this possibility.

3) RNA-seq dataset log2 of 2.

Connected to the point above, often, polysaccharide-generating genes associated with a mucoid phenotype are overexpressed in response to envelope stress (due to the lack of capsule), hence our interest in the RNA-seq dataset and the cutoff that was applied to the data described in the volcano plot Fig1c. A log2 fold change of 2, is a 4-fold change in expression, which is quite high. It seems reasonable that changes in gene expression below 4-fold could have a dramatic effect on the cell's physiology and the phenotypes associated with the grey colonies. While we applaud the inclusion of all data for all genes in the source data file, it seems possible that other gene expression changes relevant for the manifestation of the grey colony's phenotype are being missed by this high-fold change. Some acknowledgement of this, or perhaps that the relevant pulse of gene expression responsible for the phenotypes has been temporally missed by the RNA-seq approach seems appropriate.

4) Associated phenotypes of the grey colony

The authors say that cells lacking capsules are more adherent, but this is debated in the Nature Reviews article by Gao et al 2024 entitled "Bacterial capsule occurrence, mechanism, and function." The strong position of the authors, in this regard, should be dampened.

In a similar vein, the authors state in their rebuttal that "The changes in biofilm production and motility arise because the capsule is lost." This statement is misleading, because the effect could be indirect. As mentioned above, we have raised other possible reasons for the phenotypes associated with the grey colony, but these were dismissed. A softer stance is advised or experiments to support the authors' strong position are required.

Reviewer #2 (Remarks to the Author):

Reviewer #3 (Remarks to the Author):

Overall, we feel that the authors have adequately addressed our concerns. This is an impactful study that is a good fit for Nature Communications.

However, we have one remaining outstanding question. We agree that plasmids and phage are of great interest to many, and, in this manuscript, they are discussed in the context of H-NS mediated transposition to speculate on the effects of H-NS on their evolution. However, H-NS (at least in *E. coli* and *Salmonella*) is expected to target these mobile elements, so what is the reason to single them out? Of course, plasmid-encoded H-NS homologs could ruin this expectation, and perhaps the authors are working on this. We nonetheless think that comparing correlation coefficients between H-NS binding and transposition on plasmid vs chromosome could add more depth to discussion.

Reviewer #4 (Remarks to the Author):

In the revised version of their manuscript, the authors have properly addressed all of my concerns. As a suggestion, should the paper go through a new round of revision, the authors may consider placing the paragraph describing the Flag-tagged H-NS construction (lines 339-349) in the same section with the description of the insertion of ISAbA13 at the *ompW* locus (lines 360-368), perhaps creating a new section entitled "Strain construction". Also, please note that the added text contains a spelling error (the letter "k" missing in "lacking" on line 344), but this can be corrected at the proof stage.

Reviewer #5 (Remarks to the Author):

Secondly, there is no description in the discussion of StpA and how it relates to H-NS. The added sentence “this may explain why phage and plasmids encode H-NS modulating factors” only relates to their finding that H-NS2 is encoded by this strain of *A. baumannii*. But this hasn't been shared with the readers – it only make sense if you know the strain has h-ns2. The discussion would merit from some enhanced discussion of the potential/scope that StpA and other H-NS-like protein may play in the process that has been studied - transposon capture.

We agree, but our paper is not about StpA and StpA does not have relevance to the current story given that it is no encoded by our *A. baumannii* strain. One might also ask why we don't discuss any other H-NS-like protein: Hha, Ler, or the many plasmid encoded H-NS-like proteins.

In the sentence “this may explain why phage and plasmids encode H-NS modulating factors” we are not referring to H-NS-2, we are referring to the fact that there are many H-NS modulating (rather than H-NS-like, which is slightly different))proteins out there. Some of these are H-NS-like factors but some (phage encoded) are H-NS inhibitors that do not resemble H-NS. The sentence in question is not *A. baumannii* specific either.

We are happy with the current discussion but will likely get into the role of H-NS modulators in other work.

2) Grey colonies and mucoidy.

From our vantage, and likely others will agree, the grey colony's morphology looks mucoid. Additionally, other phenotypes displayed by this colony and characterized here, i.e., more adherence and less motility, are classically exhibited by mucoid cells.

If the grey colonies were mucoid, this would have been obvious during our day to day experiences working with the strain on agar plates. They are not mucoid in our hands and we have had many experiences of working with mucoid strains in the past.

In the rebuttal, the authors appear to indicate that mucoidy is only associated with strains that are producing capsule, but this is not accurate. Colonies can become mucoid even when the classic capsule genes are deleted. Instead, the mucoid phenotype manifests when cells are producing large amounts of EPS that gels on the cell's surface (see Lembre book chapter doi 10.5772/51213, “Polysaccharide molecules can interact with themselves or with heterologous ions and molecules to yield gels, often with multivalent cations playing a significant role in the process.” Taken from above citation; Introduction, paragraph 2).

We were referring to the papers that the reviewer cited as examples of mucoid *A. baumannii*. These all showed mucoidy due to capsule over production. Our colonies are not mucoid.

All it takes is for other polysaccharide-generating pathways to be upregulated. Are there other polysaccharide biosynthetic genes in the genome of AB5075? Are other sugars being produced in large quantities? For instance, in *P. aeruginosa*, mucoidy often results from an overproduction of alginate. This is why a string test was suggested in our initial review. It's a very simple assay to assess mucoidy and would address this point directly, eliminating guesses by readers about this possibility.

As noted above, it would be obvious if the colonies were mucoid, by their appearance and when picking colonies from plates. We have not seen this. Also, as mentioned below, if some other polysaccharide biosynthetic genes were being upregulated we'd see this in RNA-seq, they are not.

Last, we would like to stress that we are not the only people to have observed the grey phenotype. Nobody has associated it with mucoidy.

3) RNA-seq dataset log₂ of 2.

Connected to the point above, often, polysaccharide-generating genes associated with a mucoid phenotype are overexpressed in response to envelope stress (due to the lack of capsule), hence our interest in the RNA-seq dataset and the cutoff that was applied to the data described in the volcano plot Fig1c. A log₂ fold change of 2, is a 4-fold change in expression, which is quite high. It seems reasonable that changes in gene expression below 4-fold could have a dramatic effect on the cell's physiology and the phenotypes associated with the grey colonies. While we applaud the inclusion of all data for all genes in the source data file, it seems possible that other gene expression changes relevant for the manifestation of the grey colony's phenotype are being missed by this high-fold change. Some acknowledgement of this, or perhaps that the relevant pulse of gene expression responsible for the phenotypes has been temporally missed by the RNA-seq approach seems appropriate.

See above, other polysaccharide related genes are not upregulated. The fold change we have used is the cut-off we typically apply in our work. All cut-offs are somewhat arbitrary but the data are available if anybody wants to take a look and analyse them their own way.

4) Associated phenotypes of the grey colony

The authors say that cells lacking capsules are more adherent, but this is debated in the Nature Reviews article by Gao et al 2024 entitled "Bacterial capsule occurrence, mechanism, and function." The strong position of the authors, in this regard, should be dampened.

We strongly disagree. We do not make a general point about cells without capsule being more adherent. We talk only about the grey colonies. These definitely lack capsule and they are far more adherent. Our data in Figures 1 and S1 clearly show this unambiguously.

In a similar vein, the authors state in their rebuttal that "The changes in biofilm production and motility arise because the capsule is lost." This statement is

misleading, because the effect could be indirect. As mentioned above, we have raised other possible reasons for the phenotypes associated with the grey colony, but these were dismissed. A softer stance is advised or experiments to support the authors' strong position are required.

We disagree and, although we stated this in the rebuttal, it is not stated in the paper. In the paper we simply document the phenotypes. In fact, in cases such as natural transformation, we specifically say the effect is like to be indirect.

Again, we point out that others have linked similar phenotypes to the K-locus.

Reviewer #2 (Remarks to the Author):

Reviewer #3 (Remarks to the Author):

Overall, we feel that the authors have adequately addressed our concerns. This is an impactful study that is a good fit for Nature Communications.

However, we have one remaining outstanding question. We agree that plasmids and phage are of great interest to many, and, in this manuscript, they are discussed in the context of H-NS mediated transposition to speculate on the effects of H-NS on their evolution. However, H-NS (at least in *E. coli* and *Salmonella*) is expected to target these mobile elements, so what is the reason to single them out? Of course, plasmid-encoded H-NS homologs could ruin this expectation, and perhaps the authors are working on this. We nonetheless think that comparing correlation coefficients between H-NS binding and transposition on plasmid vs chromosome could add more depth to discussion.

I suspect we're talking past each other a little here. We single phage and plasmids out for slightly more attention because insertion into them could inactivate their mobility, which is interesting. Obviously, phage and plasmids are often bound by H-NS, as are many other chromosomal loci.

Correlation between H-NS binding and transposition on the chromosome is no different to our observations for the plasmids/phage (the latter are all chromosomally encoded anyway, so included in our correlation measurements).

We would prefer not to alter the discussion at this stage, but appreciate the comment.

Reviewer #4 (Remarks to the Author):

In the revised version of their manuscript, the authors have properly addressed all of my concerns. As a suggestion, should the paper go through a new round of revision, the authors may consider placing the paragraph describing the Flag-tagged H-NS construction (lines 339-349) in the same section with the description of the insertion of ISAb13 at the ompW locus (lines 360-368), perhaps creating a new section entitled “Strain construction”. Also, please note that the added text contains a spelling error (the letter “k” missing in “lacking” on line 344), but this can be corrected at the proof stage.

We have made the suggested change and corrected the spelling error.

Reviewer #5 (Remarks to the Author):
